# FALL3D-8.0: a computational model for atmospheric transport and deposition of particles, aerosols and radionuclides – Part 2: model validation

Andrew T. Prata[1], Leonardo Mingari[1], Arnau Folch[1], Giovanni Macedonio[2], and Antonio Costa[3]

[1]Barcelona Supercomputing Center (BSC), Barcelona, Spain
[2]Istituto Nazionale di Geofisica e Vulcanologia, Osservatorio Vesuviano, Napoli, Italy
[3]Istituto Nazionale di Geofisica e Vulcanologia, Sezione di Bologna, Bologna, Italy

**Correspondence:** Andrew Prata (andrew.prata@bsc.es)

**Abstract.** This manuscript presents model validation results for the latest version release of the FALL3D atmospheric transport model. The code has been redesigned from scratch to incorporate different categories of species and to overcome legacy issues that precluded its preparation towards extreme-scale computing. The model validation is based on the new FALL3D-8.0 test suite, which comprises a set of four real case studies that encapsulate the major features of the model; namely, the simulation of long-range fine volcanic ash dispersal, volcanic $SO_2$ dispersal, tephra fallout deposits and the dispersal and deposition of radionuclides. The first two test suite cases (*i.e.* the June 2011 Puyehue-Cordón Caulle ash cloud and the June 2019 Raikoke $SO_2$ cloud) are validated against geostationary satellite retrievals and demonstrate the new FALL3D data insertion scheme. The metrics used to validate the volcanic ash and $SO_2$ simulations are the Structure, Amplitude and Location (SAL) metric and the Figure of Merit in Space (FMS). The other two test suite cases (*i.e.* the February 2013 Mt. Etna ash cloud and associated tephra fallout deposit, and the dispersal of radionuclides resulting from the 1986 Chernobyl nuclear accident) are validated with scattered ground-based observations of deposit load and local particle grain size distributions and with measurements from the Radioactivity Environmental Monitoring database. For validation of tephra deposit loads and radionuclides we use two variants of the normalised Root-Mean-Square Error metric. We find that FALL3D-8.0 simulations initialised with data insertion consistently improve agreement with satellite retrievals at all lead times out to 48 hours for both volcanic ash and $SO_2$ simulations. In general, SAL scores lower than 1.5 and FMS scores greater than 0.40 indicate acceptable agreement with satellite retrievals of volcanic ash and $SO_2$. In addition, we show very good agreement, across several orders of magnitude, between the model and observations for the 2013 Mt. Etna and 1986 Chernobyl case studies. Our results, along with the validation datasets provided in the publicly available test suite, form the basis for future improvements to FALL3D (versions 8 or later) and also allow for model inter-comparison studies.

## 1 Introduction

FALL3D-8.0 is the latest major version release of FALL3D (Costa et al., 2006; Folch et al., 2009), an open-source code with a 15+ year track record and a growing number of users in the volcanological and atmospheric science communities. A

companion paper (Folch et al., 2020) details the physics and the novel numerical implementation of the code, which has been redesigned and rewritten from scratch in the framework of the EU Center of Excellence for Exascale in Solid Earth (*ChEESE*).

From the point of view of model physics, a relevant improvement in the new version v8.x has been the generalisation of the code to deal with atmospheric species other than tephra including other types of particles (*e.g.* mineral dust), gases and radionuclides (see Table 3 in Folch et al., 2020, for details). These different categories and sub-categories of species can be simulated using independent sets of bins that allow for dedicated parameterisations for physics, emissions (source terms) and interactions among bins (*e.g.* aggregation, chemical reactions, radioactive decay *etc*). In FALL3D, "tephra" species are

subdivided into four subcategories, depending on particle diameter, $d$ (Folch et al., 2020): (i) fine ash ($d \leq 64$ μm), (ii) coarse ash ($64$ μm $< d \leq 2$ mm), (iii) lapilli ($2$ mm $< d \leq 64$ mm), and (iv) bomb ($d > 64$ mm). In terms of model performance, the new model version contains a much more accurate and less diffusive solver, as well as a better memory management and parallelisation strategy that notably outperforms the scalability and the computing times of the precedent code versions v7.x (Folch et al., 2020). This paper complements the Folch et al. (2020) companion paper by presenting a detailed set of real-case

validation examples, all included in the new test suite of the code. This manuscript also contains some novel aspects regarding geostationary satellite detection and retrieval of volcanic ash (Appendix A) and $SO_2$ (Appendix B), as well as the FALL3D-8.0 data insertion methodology. To furnish the initial model condition required for data insertion, the satellite retrievals are collocated with lidar measurements of cloud-top height and thickness from the CALIPSO (Cloud-Aerosol Lidar and Infrared Pathfinder Satellite Observation) platform (Winker et al., 2009). The data insertion scheme is a preliminary step towards model

data assimilation using ensembles, a novel functionality currently under development.

The manuscript is organised as follows. Section 2 provides an overview of the model test suite, detailing the file structure and contents for each of the four case studies considered for validating and testing FALL3D-8.0. Section 3 provides a description of each of the events making up the test suite, which include simulations of the June 2011 Puyehue-Cordón Caulle ash cloud, the June 2019 Raikoke $SO_2$ cloud, the February 2013 Mt. Etna ash cloud and associated tephra fallout deposit, and the dispersal

of radionuclides resulting from the 1986 Chernobyl nuclear accident. The datasets used for validation and the FALL3D model configurations used in each case are also contained in Section 3. Section 4 describes the validation metrics, which include the Structure, Amplitude and Location (SAL) metric (Wernli et al., 2008) and the Figure of Merit in Space (FMS; Galmarini et al., 2010; Wilkins et al., 2016) to quantitatively compare model results with satellite retrievals, along with the Root-Mean-Square Error (RMSE) for validation of the ground deposit simulations of tephra and radionuclides. Section 5 presents a detailed

discussion of the validation results for the four test suite cases. Finally, Section 6 presents the conclusions of the manuscript and outlines the next steps in terms of model development and applications.

## 2  The *FALL3D-8.0* test suite

FALL3D-8.0 includes both a benchmark and a test suite. The benchmark suite consists of a series of non-public small-case tests used by model developers for model verification and code performance analysis. These benchmark cases are typically

passed when substantial model updates in the master branch of the code are committed to the repository and, essentially,

contain idealised 1D or 2D cases with known analytical solutions for verification purposes (see, e.g., Figures 3, 4, and 5 in Folch et al., 2020). In contrast, the test suite includes larger-size, real-case simulations aimed at model validation. Model users can download the public test suite repository files to run the model and to check whether it has been properly installed and configured on their local machines. This paper presents the four cases from the FALL3D-8.0 test suite listed in Table 1; namely, **Puyehue-2011** (simulation of the June 2011 Puyehue-Cordón Caulle ash cloud), **Raikoke-2019** (simulation of the June 2019 Raikoke $SO_2$ cloud), **Etna-2013** (simulation of the 23 February 2013 Mt. Etna ash cloud and related tephra fallout deposit), and **Chernobyl-1986** (simulation of the dispersal and deposition of radionuclides resulting from the April 1986 Chernobyl nuclear accident). Note that the names of the validation cases are shown in bold throughout this paper.

The FALL3D-8.0 test suite repository contains independent folders (one per validation case). All case folders have the same sub-folder structure:

– *InputFiles*. This sub-folder contains all the necessary input files to run the case. The only exception is meteorological data because it typically involves substantially large files (∼10s of GB) that make the storage and the transfer to/from the GitLab public repository unpractical/unfeasible.

– *Utils/Meteo*. This sub-folder contains all the necessary files to obtain meteorological data depending on the meteorological driver (for possible options see Table 12 in Folch et al., 2020). For global datasets such as the ERA5 dataset (Copernicus Climate Change Service (C3S), 2017) used in the **Puyehue-2011** case, Python and shell scripts are provided so that the user can download and merge meteorological data consistently with the *SetDbs* pre-process task (see Section 5 in Folch et al., 2020). For mesoscale datasets (*e.g.* the WRF-ARW dataset used in the **Etna-2013** case), the sub-folder includes the meteorological model "namelists" and scripts to download the global data driving the corresponding mesoscale model simulation.

– *Utils/Validation*. This sub-folder contains all the necessary files to validate the *FALL3D-8.0* model execution results, including a file with the expected validation metrics results.

Tables 2 and 3 list the files in the test suite for the **Puyehue-2011** (Sec. 3.1) and the **Etna-2013** (Sec. 3.3) cases, respectively. The **Raikoke-2019** (Sec. 3.2) and **Chernobyl-1986** (Sec. 3.4) filenames look very similar to the **Puyehue-2011** and **Etna-2013** filenames and are not explicitly shown here. In all cases, the test suite contains the necessary information to reproduce all of the simulation and validation results shown in the present study.

## 3  Validation cases

### 3.1  Puyehue-2011

The Puyehue-Cordón Caulle volcanic complex (PCCVC), located in the southern volcanic zone of the central Andes, comprises a 20 km long, NW-SE oriented fissure system (Cordón Caulle) and the Puyehue stratovolcano (Elissondo et al., 2016). On 4 June at around 14:45 LT (18:45 UTC), a new vent opened at 7 km NNW from the Puyehue volcano (Collini et al., 2013),

initiating a remarkable example of a long-lasting plume with complex dynamics, strongly influenced by the interplay between eruptive style, atmospheric winds and deposit erosion (Bonadonna et al., 2015). The initial explosive phase of the eruption (4-14 June) was characterised by the development of eruption columns with heights oscillating between 6–14 km above sea level (a.s.l.), which correspond approximately to 4–12 km above the vent. Plume heights progressively decreased (4-6 km a.s.l) between 15 and 30 June and low intensity ash emission persisted for several months (Elissondo et al., 2016). Due to the predominant westerly winds, ash was transported towards Argentina and a wide area of the arid and semi-arid regions of northern Patagonia was severely affected by tephra dispersal and fallout. The PCCVC event stands as one of the most extraordinary examples of long-range fine ash transport observed by satellite and is the reason for selecting it as a validation case in the present study.

### 3.1.1 Validation dataset

In order to validate FALL3D-8.0 simulations of dispersal of fine ash and to test the new volcanic ash data insertion scheme, we use the retrieval method of Prata and Prata (2012) to derive fine ash mass loading estimates based on InfraRed (IR) measurements made by the SEVIRI (Spin Enhanced Visible and Infrared Imager; Schmetz et al., 2002) instrument (onboard Meteosat-9) during the 2011 Puyehue-Cordón Caulle eruption in Chile. For IR wavelengths, SEVIRI samples the Earth's full disk every 15 minutes with a spatial resolution of 3 km × 3 km at the sub-satellite point. After applying the Prata and Prata (2012) retrieval algorithm to SEVIRI at its native resolution, we resample the data onto a regular latitude-longitude grid of $0.1° × 0.1°$ (using nearest neighbour resampling), consistent with the FALL3D output grid. The satellite observations we consider for the Puyehue case study cover the time period from 00:00 UTC on 5 June 2011 to 00:00 UTC on 10 June 2011 and have a temporal resolution of 1 h (121 time steps). Details of the retrieval algorithm implementation and specific ash detection thresholds adopted for the Puyehue case study are provided in Appendix A.

### 3.1.2 Model setup

To simulate long-range, fine volcanic ash ($d \leq 64$ μm) transport from the Puyehue-Cordón Caulle eruption we carried out FALL3D runs in two configurations. The first configuration (listed as **Puyehue-2011** (A) in Table 1) was initialised at 21:00 UTC on 4 June 2011 for a duration of 99 h with a continuous emission source term. The source term was defined as a uniform distribution (HAT option) with a column height of 11.2 km a.s.l. (9 km above vent level) and 2 km column thickness. The second configuration (listed as **Puyehue-2011** (B) in Table 1) demonstrates the new data insertion scheme which was recently introduced in FALL3D-8.0 as described in Folch et al. (2020). The data insertion run was initialised at 15:00 UTC on 5 June 2011, which coincides with a CALIPSO overpass that intersected the nascent ash plume (see Fig. A4b). To constrain the vertical distribution of ash in FALL3D, we collocated the CALIOP (Cloud-Aerosol Lidar with Orthogonal Polarization) total attenuated backscatter profiles with ash-affected SEVIRI pixels. Note that the vertical distribution is only required for the data insertion time. Based on these coincident observations we set the cloud-top height and thickness of the data-inserted ash cloud to 13 km a.s.l. and 2 km, respectively. In addition, to account for ash erupted after the data insertion time, we initialised the same source term specified in the **Puyehue-2011** (A) configuration (i.e. continuous, uniform source emission up to 11.2 km

a.s.l.) at the data insertion time (15:00 UTC on 5 June 2011). For both model configurations, the meteorological dataset used was ERA5 reanalysis (Copernicus Climate Change Service (C3S), 2017), which has a horizontal spatial resolution of 0.25° and 137 model levels. The output domain size in both cases was set to 15–75°S and 20–55°W (0.1° latitude-longitude grid resolution). Fine ash species were considered for both model configurations to simulate airborne PM10 mass loadings. Further details of both model configurations are summarised in Table 1.

## 3.2 Raikoke-2019

On 21 June 2019, a small island volcano, Raikoke (48.292° N, 153.25° E, 551 m a.s.l.), underwent a significant explosive eruption disrupting major aviation flight routes across the North Pacific. Raikoke is located in the central Kuril Islands, a remote island chain that lies South of Russia's Kamchatka peninsula. Ground-based networks are sparse in this area and so satellite observations were crucial for tracking the volcanic ash and $SO_2$ produced by the eruption. The eruption sequence was characterised by a series of ∼9 'pulses', injecting ash and gases into the atmosphere. The International Space Station captured a unique view of the eruption's umbrella plume during its initial explosive phase which was reminiscent of the 2009 Sarychev Peak umbrella plume (https://earthobservatory.nasa.gov/images/145226/raikoke-erupts). The eruption sequence was captured extremely well by the Japanese Meteorological Agency's Himawari-8 satellite at both IR and visible wavelengths. According to our analysis of the satellite data, the initial explosive phase began at around 18:00 UTC on 21 June (05:00 LT on 22 June at around sunrise) and ended at around 10:00 UTC on 22 June (21:00 LT just before sunset). The Smithsonian Institution's Global Volcanism Program (GVP) report on the 2019 Raikoke eruption also documents less intense activity at the volcano from 23–25 June following the initial explosive phase (Global Volcanism Program, 2019). During the initial explosive phase on 21 June 2019, a significant amount of $SO_2$ was injected into the atmosphere making the eruption an ideal case to study long-range $SO_2$ transport. Preliminary analysis of TROPOMI $SO_2$ retrievals indicated that ∼1.4–1.5 Tg $SO_2$ was injected into the atmosphere (Global Volcanism Program, 2019). Hyman and Pavolonis (2020) present $SO_2$ retrievals based on Cross-track Infrared Sounder (CrIS) measurements and show a time-series of $SO_2$ total mass with a peak between 1–1.1 Tg $SO_2$. These $SO_2$ mass estimates suggest that the Raikoke eruption resulted in the largest injection of $SO_2$ into the atmosphere since Nabro (4.5 Tg $SO_2$) in 2011 (see Carn et al., 2016, Table 2 for a list of major $SO_2$ mass injections recorded by satellite).

### 3.2.1 Validation dataset

Existing volcanic $SO_2$ datasets are mainly based on polar-orbiting IR (e.g. Realmuto et al., 1994; Watson et al., 2004; Prata and Bernardo, 2007; Corradini et al., 2009; Clarisse et al., 2012; Carboni et al., 2016) and UV (e.g. Yang et al., 2007; Carn et al., 2015; Theys et al., 2017) satellite observations. However, in order to validate FALL3D at high temporal resolution (hourly or better), geostationary satellite observations seem preferable. To our knowledge, no operational geostationary $SO_2$ products are currently available. Therefore, in order to validate the new $SO_2$ scheme in FALL3D-8.0, we apply the three-channel $SO_2$ retrieval proposed by Prata et al. (2003) to the Advanced Himawari Imager (AHI) instrument aboard the Himawari-8 geostationary satellite (Bessho et al., 2016). We selected the AHI due its exceptional spatial and temporal coverage during the Raikoke eruption. For IR wavelengths, AHI samples the earth's full disk every 10 minutes with a spatial resolution of 2 km

× 2 km at the sub-satellite point. As we did for the SEVIRI ash retrievals (*cf.* Sect. 3.1.1), we resampled the AHI data from its native resolution to a regular grid (spatial resolution of 0.1° × 0.1°) and output the retrievals at 1 h temporal resolution

for comparison with FALL3D simulations. The observational time period considered for the Raikoke case is from 18:00 UTC on 21 June 2019 to 18:00 UTC on 25 June 2019 (97 time steps). Details of the implementation of the Prata et al. (2003) retrieval scheme and methods used to detect volcanic $SO_2$ for the Raikoke eruption are provided in Appendix B. The retrievals presented here indicate a maximum total mass of ∼1.4 Tg $SO_2$, which is in broad agreement with other satellite-based $SO_2$ mass estimates (∼1-1.5 Tg $SO_2$; Global Volcanism Program, 2019; Hyman and Pavolonis, 2020). The present $SO_2$ retrieval

scheme was originally developed for HIRS/2 (High-resolution Infrared Radiation Sounder) data. The error budget from Prata et al. (2003) suggests that errors from 10-20% are to be expected for detectable $SO_2$ column loads up to 800 DU. We expect that the Himawari-8 retrieval errors will be of similar magnitude or better.

### 3.2.2   Model setup

For volcanic $SO_2$ dispersion from the Raikoke eruption, the test suite considers two simulations to validate the new $SO_2$ option

in FALL3D in the same fashion as the volcanic ash simulations (*i.e.* simulations with and without data insertion). The model configuration without data insertion (listed as **Raikoke-2019** (A) in Table 1), was initialised at 18:00 UTC on 21 June 2019, which coincides with the onset of the eruption observed by the Himawari-8 satellite (Sect. 3.2.1). For the $SO_2$ source term, we use a Suzuki distribution and a variable column height (between 3.5 and 15.5 km a.s.l., which corresponds to 3–15 km above vent level) with a source duration of 14 h and total run time of 72 h (Table 1). The data insertion model configuration

(listed as **Raikoke-2019** (B) in Table 1) was initialised at 18:00 UTC on 22 June 2019 (1 day after the beginning of the eruption) as this is a time when the $SO_2$ cloud was completely detached from source (Fig. 4a). Based on several collocations of CALIOP attenuated backscatter profiles with $SO_2$-affected AHI pixels (*e.g.* Fig. B3), we set the $SO_2$ cloud-top height and thickness to 13.5 km and 2.5 km, respectively, at the data insertion time. We note that CALIOP total attenuated backscatter measurements are not sensitive to $SO_2$ gas and so we make the assumption that $SO_4^{2-}$ aerosols were collocated with $SO_2$ in

order to assess the likely heights and thicknesses of the Raikoke $SO_2$ clouds (Carboni et al., 2016; Prata et al., 2017). For both model configurations, the meteorological dataset used was GFS forecast (NCEP, 2015), which has a horizontal spatial resolution of 0.25° and 34 pressure levels. The output domain size in both cases was set to 35–65°N and 150–210°E (0.1° latitude-longitude grid resolution). More details of the $SO_2$ model configurations can be found in Table 1.

### 3.3   Etna-2013

On 23 February 2013 the eruptive activity of Mt. Etna increased significantly and at 18:15 UTC a buoyant plume rose up to 9 km a.s.l. (5.68 km above the vent) along with incandescent lava fountains exceeding 500 m above the crater (Poret et al., 2018). The resulting ash plume extended towards the NE for more than 400 km away from the source and moderate ash fallout was observed throughout the Italian regions of Calabria and Puglia. Due to the extensive field work carried out following the 2013 Etna eruption, it is an ideal case for validating simulations of tephra ground deposits.

### 3.3.1 Validation dataset

To validate tephra deposition from the 2013 Mt. Etna eruption, we use the ground deposit observations reported by Poret et al. (2018), which consist of measurements of mass loading per unit area and local Grain-Size Distribution (GSD) obtained by sieve analysis (except for the farthest sample, which required a different experimental technique for the measurement of GSD). Samples were collected a few hours after the volcanic eruption at 10 different locations (S1–S10). Locations of sampling sites S7–S10 are indicated in Fig. 6. Proximal sites (S1–S7) are located between 5 and 16 km from the vent, whereas the rest of samples (S8–S10) correspond to the locations of Messina (Sicily, S8), Cardinale (Calabria, S9) and Brindisi (Puglia, S10), the latter located at about 410 km from the volcano.

### 3.3.2 Model setup

In order to simulate tephra deposition from the Mt. Etna eruption, we use high-resolution wind fields generated from the ARW (Advanced Research WRF) core of the WRF (Weather Research and Forecasting) model (Skamarock et al., 2008) on a single-domain configuration consisting of $700 \times 700$ grid points with horizontal resolution of 4 km and 100 vertical levels with a maximum height of 50 hPa. The initial and boundary conditions for WRF/ARW were extracted from hourly ERA5 reanalysis data (Copernicus Climate Change Service (C3S), 2017), which has a spatial resolution of $0.25°$ and 137 vertical model levels. The FALL3D run was initialised with a start time of 18:00 UTC on 23 February and with a uniform source distribution (HAT option) reaching 5.5 km above the vent (*i.e.* 8.7 km a.s.l) and a thickness of 3.5 km. We considered a constant mass flow rate ($3.814 \times 10^5$ kg s$^{-1}$) between 18:15 and 19:18 UTC on 23 February 2013. The model was configured with a horizontal resolution of $0.015°$ and 60 vertical levels up to 11 km in a computational domain comprising all deposit sampling locations. The particle Total Grain Size Distribution (TGSD) was discretised into 32 bins with diameter in $\Phi$ units in the range $-6\Phi \leq d \leq 6\Phi$ (diameter can be expressed in milimeters through the relationship $d = 2^{-\Phi}$), densities varying between 1000 and 2500 kg m$^{-3}$ for coarser and finer bins respectively, and a constant sphericity of 0.92. This test case considers a bi-Gaussian (in $\Phi$) TGSD defined as:

$$f(\Phi) = \frac{p}{\sigma_c \sqrt{2\pi}} \exp\left(-\frac{(\Phi - \mu_c)^2}{2\sigma_c^2}\right) + \frac{1-p}{\sigma_f \sqrt{2\pi}} \exp\left(-\frac{(\Phi - \mu_f)^2}{2\sigma_f^2}\right) \tag{1}$$

where $\mu_f$ and $\sigma_f$ are the mean and standard deviation for the fine subpopulation, $\mu_c$ and $\sigma_c$ are the mean and standard deviation for the coarse subpopulation, and $p$ and $1-p$ the relative weight of each subpopulation. Poret et al. (2018) performed numerical simulations using $p = 0.59$, subpopulation means of $\mu_c = -2.96$, $\mu_f = 0.49$, and standard deviations of $\sigma_c = 1.03$, $\sigma_f = 0.79$, for coarse and fine populations respectively. However, as already noted by Poret et al. (2018), such a TGSD underestimates the fine ash fraction. In order to correct this drawback when using a bi-Gaussian TGSD, the **Etna-2013** test case was run considering a finer TGSD having $\mu_c = -2.96$, $\sigma_c = 1.03$, $\mu_f = 2.54$, $\sigma_f = 0.38$, and $p = 0.7$ (note that the latter parameters were calibrated through a trial-and-error procedure). Table 1 summarises the rest of model configuration options and the final tephra ground load map for a simulation time of 10 h is shown in Fig. 6.

### 3.4 Chernobyl-1986

One of the most serious nuclear accidents on Earth occurred on 25 April 1986 at 21:23 UTC at the Chernobyl Nuclear Power
Plant (NPP) in Ukraine. The radioactive material released as a consequence of two explosions at the NPP was transported by
atmospheric winds thousands of kilometers away from the source, resulting in European-wide dispersal of several radionuclide
isotopes. The availability of the Radioactivity Environmental Monitoring (REM) database (De Cort et al., 2007) along with
the simulation results of Brandt et al. (2002) make the 1986 Chernobyl nuclear accident a good case study to validate the new
radionuclide scheme in FALL3D-8.0.

#### 3.4.1 Validation dataset

To validate FALL3D-8.0 radionuclide simulations of the 1986 Chernobyl nuclear accident, the test suite considers the dataset
reported by Brandt et al. (2002), who used high quality deposition measurements from the REM database (De Cort et al.,
2007). The REM data used for validating the **Chernobyl-1986** case are provided in the test suite public repository.

#### 3.4.2 Model setup

The ability to simulate the transport and deposition of radionuclides is a new feature in FALL3D-8.0. In order to simulate the
dispersion of radioactive material, estimations of the emission rate and of the particle size distributions and settling velocities
of radioactive material (*e.g.* $^{134}$Cs, $^{137}$Cs, and $^{131}$I isotopes) are needed. Unfortunately, existing estimations of such parameters
are uncertain due to obvious in-situ measurement difficulties and to the interaction of isotopes with other atmospheric particles
and aerosols (Brandt et al., 2002), and rely on reconstructions and/or on best-fitting the available measurements in the region
of interest at the time of the accident. For example, on the basis of high quality deposition measurements from the REM
database (De Cort et al., 2007), Brandt et al. (2002) reconstructed the source term and identified two emission phases with
different vertical mass distributions. The **Chernobyl-1986** test suite case uses the source term reported by Brandt et al. (2002),
parameterized using the SUZUKI and HAT options in FALL3D (see Table 1 for more details of the input parameters). On
the other hand, effective settling velocities typically range from 0.5 to 5 $\mathrm{mm\,s^{-1}}$ for $^{137}$Cs and from 1 to 20 $\mathrm{mm\,s^{-1}}$ for $^{131}$I
(Brandt et al., 2002). Considering these ranges and discretising velocities into four classes (see Table 5) we chose the effective
classes and fractions for **Chernobyl-1986** based on the best fit of the simulations with the measured deposit radioactivity on 10
May 1986. The best fit was performed by changing the nuclide size distribution (each class having a different settling velocity)
and keeping the total mass constant. Minimization was performed through a regular grid search varying the relative weights of
the nuclide size classes between 0 and 1 at steps of 0.01. The **Chernobyl-1986** case considers the computational domain shown
in Fig. 8 for the period from 24 April to 10 May 1986, considering the input values reported in Table 5 and the meteorological
fields obtained from ERA5 reanalysis, which accounts for atmospheric diffusion, wet deposition and radioactive decay.

## 4 Validation metrics

### 4.1 Structure, Amplitude and Location

Both qualitative and quantitative validation approaches have been used to validate previous versions of FALL3D against satellite observations (e.g. Corradini et al., 2011; Folch et al., 2012). The **Puyehue-2011** and **Raikoke-2019** test suite cases use the SAL metric (Wernli et al., 2008) to quantitatively compare satellite retrievals of volcanic ash and $SO_2$ to the corresponding simulations with and without data insertion. The SAL metric was originally developed for validation of precipitation forecasts against radar and satellite data (Wernli et al., 2008). However, Dacre (2011) demonstrated its use for validation of air pollution simulations and Wilkins et al. (2016) employed the SAL for dispersion model validation against IR satellite volcanic ash retrievals for the 2010 Eyjafjallajökull eruption. More recently, the SAL has also been used to compare online vs. offline model simulations of volcanic ash (Marti and Folch, 2018). As in Wilkins et al. (2016) and Marti and Folch (2018), we also use the FMS score (Figure of Merit in Space) as a complement to SAL for comparing the spatial coverage of observed vs. modelled fields. A detailed mathematical description of the SAL metric is presented in Wernli et al. (2008) and so we only provide a brief description of each of the components of SAL (i.e. S, A and L) in the following subsections. The main requirement for calculating SAL is the determination of model and observation *objects*. Objects are identified as clusters of contiguous pixels whose magnitude is above some threshold corresponding to a physical quantity determined from observations. In our case, the threshold is determined based on the detection limit of the satellite retrievals. For the SEVIRI ash retrievals (**Puyehue-2011** test suite case), we used a threshold of $0.2 \, \mathrm{g \, m^{-2}}$ consistent with the threshold suggested by Prata and Prata (2012). For the AHI $SO_2$ retrievals (**Raikoke-2019** test suite case), there is currently no commonly accepted detection threshold. For the purposes of identifying $SO_2$ objects, we allowed for a threshold of 5 DU, noting that the minimum detected $SO_2$ total column burdens at each time step in the satellite retrievals were generally $\sim$8–10 DU. After identifying objects for both the observation (satellite retrievals) and model fields, we compute the SAL as the sum of the absolute values of S, A and L, which results in an index that varies from 0 (best agreement) to 6 (worst agreement). All comparisons between observations and model simulations were made using a regular $0.1° \times 0.1°$ latitude-longitude grid.

#### 4.1.1 Amplitude

The Amplitude (A) metric is the simplest of the three SAL components and compares the normalised difference of the mass-averaged values of the observation and model fields. It can vary from -2 to +2 where negative (positive) values indicate that the model is under-predicting (over-predicting) the mass when compared with observations.

#### 4.1.2 Location

The Location (L) metric has two components ($L = L_1 + L_2$). $L_1$ is calculated as the distance between the centers of mass between the model and observation objects, normalised by the maximum distance across the specified domain. It can vary from 0 to +1 and is considered a first-order indication of the accuracy of the model simulation compared with observations.

However, $L_1$ can equal 0 (suggesting perfect agreement) for situations where observation and model fields clearly do not agree. For example, Wernli et al. (2008) describe the case of two objects at opposite sides of the domain having the same center of mass as a single object in the center of the domain. $L_2$ was introduced to handle these situations by considering the weighted average distance between the overall center of mass and the center of mass of each individual object for both model and observation fields. $L_2$ is computed by taking the normalised difference between the model weighted average distance and the observation weighted average distance. It is scaled such that it varies from 0 to +1 (to vary over the same range as $L_1$), meaning that L varies in the range from 0 to +2.

### 4.1.3 Structure

The Structure (S) metric is the most complex of the three metrics used to construct SAL. The general idea is to compute the normalised 'volume' of all individual objects for each dataset (i.e. the model and observation fields). The normalised volumes are computed by dividing the total (sum) mass of each object by its maximum mass. The weighted mean of the normalised volumes is then computed for the observation and model fields and S is computed by taking the difference between the weighted means. The S metric can vary from -2 to +2, where negative values indicate that modelled objects are too small or too peaked (or a combination of both) compared to the observed fields.

## 4.2 Figure of Merit in Space

The FMS score compares the spatial coverage of observed vs. modelled fields. It is simply the area of intersection divided by the area of union between the ash mass loading observation and model fields:

$$\mathrm{FMS} = \frac{A_{\mathrm{mod}} \cap A_{\mathrm{obs}}}{A_{\mathrm{mod}} \cup A_{\mathrm{obs}}}, \tag{2}$$

where $A_{\mathrm{mod}}$ and $A_{\mathrm{obs}}$ are the modelled and observed ash mass loading areas, respectively. The FMS varies from 0 (no intersection) to 1 (perfect overlap).

## 4.3 Root-Mean-Square Error

The **Etna-2013** and **Chernobyl-1986** test suite cases use the Root-Mean-Square Error (RMSE) metric to quantitatively compare observations and numerical simulations. For this purpose, we consider two variants of the normalised RMSE, defined as (Poret et al., 2018):

$$RMSE_j = \sqrt{\sum_i^N w_j (\mathrm{MOD}_i - \mathrm{OBS}_i)^2} \qquad \text{with} \qquad w_1 = \frac{1}{\sum_i^N \mathrm{OBS}_i^2} \qquad w_2 = \frac{1}{N}\frac{1}{\mathrm{OBS}_i^2} \tag{3}$$

where $j = 1, 2$ depends on the RMSE choice, $w_j$ refers to the weighting factor used to normalize RMSE, and $\mathrm{OBS}_i$ and $\mathrm{MOD}_i$ are the observed and simulated mass loading related to the $i$-th sample over a set of $N$ samples. The weights correspond to different assumptions about the error distribution (e.g. Aitken, 1935; Costa et al., 2009). The $\mathrm{RMSE}_1$ calculated with $w_1$ refers

to a constant absolute error, whereas the RMSE$_2$, calculated using a proportional weighting factor $w_2$, considers a constant relative error (e.g. Bonasia et al., 2010; Folch et al., 2010; Poret et al., 2018).

## 5   Validation results

### 5.1   Puyehue-2011

Figures 1 and 2 compare satellite retrievals and model simulations for the **Puyehue-2011** case at the data insertion time (15:00 UTC on 5 June 2011) as well as 24, 48 and 72 h after the insertion time for runs with and without data insertion, respectively. The time-series of each validation metric are also shown in Fig. 3a,c and summarised in Table 4. Comparison of Figs. 1a and 2a highlight the advantage of a data insertion scheme to specify model initial conditions. Note that for simulations with data insertion at the data insertion time the validation metrics reflect perfect scores (*i.e.* SAL = 0 and FMS = 1). For the simulation

without data insertion (Fig. 2a), the plume has already begun to deviate from the satellite observations with too much mass dispersing towards the south. This is reflected in both the SAL score of 1.93 and FMS score of 0.22 at this time. The data insertion scheme (Fig. 1a) naturally corrects for this by taking advantage of good quality satellite observations of the vertical and horizontal structure of the Puyehue ash plume at the insertion time. For the data insertion simulations, FALL3D accurately represents the spatial structure of the satellite retrievals after 24 hours with a SAL score of 1.3 and FMS of 0.42 (Fig. 1b; see

Supplementary Material for the full animation of the data insertion simulations). In addition, the accuracy of the simulations over the first 24 hours shows a marked improvement when compared to the simulations without data insertion (Fig. 2b; SAL = 1.84; FMS = 0.32). The validation metric time-series show this in more detail (Figs. 3a and c). For the simulations without data insertion, the SAL score remains above 2 for most of the first 24 hours while the SAL gradually increases from 0 to 1.3 for the simulation with data insertion. Comparison of Figs. 1b and 2b shows that the data insertion simulation is better able to

capture the northern portion of the plume than the simulation without data insertion at this time (an increase in the FMS by 0.1). Inspection of the time-series of the individual validation metrics for the simulations with data insertion (Fig. 3a) reveals that the SAL is largely being affected by increases in the L metric (i.e. increases in the distance between the centres mass between the observation and model fields) and decreases in the A metric (model under-predicting mass compared to observations). The S metric only exhibits minor deviations when compared to the observations during the first 24 hours after data insertion. At 48

330   hours, the simulations with and without the data insertion are almost identical (minor differences in the modelled ash contours near 30° S, 15° W). This is because, at this time, almost all of the ash used in the data insertion has exited the domain. For the simulations without data insertion, the SAL score is 1.2 and FMS is 0.14 (Fig. 2c; Table 4); however, at around 36 hours the SAL reached above 2 and then decreased sharply (Fig. 3c). The reason for the sudden reduction in SAL just after 36 hours is most likely due to the satellite retrievals being compromised by cloud interference at this time in addition to the continual

input of mass due to the emission source in the model simulations. Note that this input of mass was included to account for ash erupted after the data insertion time. The satellite retrievals capture some of the ash plume near source (Figs. 1c and 2c), but cannot be expected to accurately characterise the plume at this location due to its high opacity in the IR window. Another difference between the model and observations at this time is the large difference in the centres of mass (L = 0.32). This is due

to the high mass loadings near source in the model fields and high mass loadings near the centre of the domain (43° S, 35° W)
in the observed fields. The satellite is likely over-estimating mass in this part of the ash cloud because of underlying water/ice clouds that have not been accounted for in the radiative transfer modelling. Simulations with and without data insertion are identical after 72 hours as all ash used in the data insertion scheme has exited the domain (Fig. 1d and 2d). At this time, the SAL has reached a score of 1.71 and the FMS has decreased to 0.10, reflecting a model predictive performance degrading over time.

## 5.2 Raikoke-2019

Figures 4 and 5 show a comparison between the satellite retrievals and model simulations with and without data insertion, respectively, in addition to the SAL and FMS validation metrics. The time-series of validation metrics for the **Raikoke-2019** case are shown in Figs. 3b and 3d and are summarised in Table 4. The AHI retrievals of the $SO_2$ plume at the beginning of the eruption (Fig. B3a) were likely compromised by interference of ice particles in the initial eruption plume (Prata et al., 2003;
Doutriaux-Boucher and Dubuisson, 2009). In addition, retrievals early on in the plume's dispersion may have been affected by band saturation caused by extremely high $SO_2$ column loads. At the data insertion time (22 June 2019 at 18:00 UTC), the SAL score for the $SO_2$ simulation without data insertion is 2.87 and the FMS is 0.32. Therefore, applying data insertion at this time represents a significant correction to the model simulation when compared against the satellite retrievals (compare Fig. 4a and Fig. 5a). The main difference between the satellite retrievals and simulation without data insertion is that the model indicates
a portion of the $SO_2$ plume connecting back to the volcano while this feature is not present in the observations. TROPOMI observations of the $SO_2$ cloud confirm this spatial structure (see Fig. 13 of Global Volcanism Program, 2019) and highlight the importance of understanding the limitations of the AHI retrievals used for data insertion in the present study. The reason for the lack of detection of $SO_2$ in this region in the AHI retrievals is probably due to water vapour interference, implying that this part of the plume was at lower altitudes than the main $SO_2$ cloud. Indeed, $SO_2$ height retrievals from CrIS data show that
plume heights varied from ~3–7 km a.s.l. in this region (see Fig. 5 of Hyman and Pavolonis, 2020).

For the simulations without data insertion, at 24 hours after insertion, the validation metrics exhibit minor changes with SAL decreasing from 2.87 to 2.59 and the FMS from 0.32 to 0.23 (Fig. 5a, b). For the simulations with data insertion, SAL has steadily increased from 0 to 1.21 while the FMS has decreased from 1 to 0.29 over the first 24 hours (Fig. 4a, b; see Supplementary Material for the full animation of the data insertion simulations). Figure 3b shows that the SAL score for
the simulation with data insertion is mostly affected by the S and A scores whereas the L score is low (0.05) indicating that FALL3D is able to track the centre of mass of $SO_2$ very well when initialised with satellite retrievals (Fig. 3b). In this case the A metric is negative, meaning that the model is under-predicting the mass when compared to the satellite retrievals. Given that FALL3D-8.0 does not include $SO_2$ chemistry, and only includes $SO_2$ deposition mechanisms (Folch et al., 2020), it is unlikely that the model is losing $SO_2$ mass too rapidly. Instead, this under-prediction is probably being driven by the observed increase
in mass retrieved by the satellite after the insertion time, which cannot be accounted for in the data insertion scheme if no new sources of $SO_2$ are included in the model simulations. An increase in mass, even after the $SO_2$ cloud has detached from source, could be related to the detection sensitivity of the satellite retrieval and the influence of water vapour (Prata et al., 2003;

Doutriaux-Boucher and Dubuisson, 2009). For example, if the $SO_2$ cloud is in a region of high water vapour initially and then moves into a drier region it is possible that more $SO_2$ will be detected thus increasing the retrieved mass. This effect could also occur if the $SO_2$ layers are transported vertically in the atmosphere. Another interesting mechanism for the observed increase in $SO_2$ mass after the plume has detached from source is ice-$SO_2$ sequestration. This phenomenon occurs when ice particles in the nascent plume sequester $SO_2$ initially and then release it later on as the ice sublimates in the UTLS (e.g. Rose et al., 2001, 2003, 2004; Guo et al., 2004; Prata et al., 2007; Fisher et al., 2019).

At 48 hours, for the simulations without data insertion (Fig. 5c), the SAL score actually improves (decreasing from 2.58 to 1.88) and the FMS largely remains the same (decreasing from 0.23 to 0.20). The improvement in the SAL score can be attributed to a steady increase in S metric and decrease in the A metric (Fig. 3d). This indicates that the structure and mass (amplitude) simulated without data insertion are converging towards that observed by the satellite over 48 hours. For the simulations with data insertion, at 48 hours after insertion (Fig. 4c), the SAL score has continued to increase (from 1.21 to 1.38) and the FMS has continued to decrease (from 0.29 to 0.25). In general, at all lead times, the validation metrics indicate that the data insertion simulations provide better agreement with observations than the simulations without data insertion (Table 4).

### 5.3 Etna-2013

Figure 7a compares modelled and observed tephra loading for the 2013 Mt. Etna eruption at all observation sites (detailed in Sect. 3.3.1). Note that all points are found within a factor 3 around the perfect agreement line (black solid line) across four orders of magnitude (from $10^{-3}\,\mathrm{kg\,m^{-2}}$ to more than $10\,\mathrm{kg\,m^{-2}}$). This good agreement is also observed at a bin level in the local grain size distributions. According to the model predictions presented here, unimodal local distributions (negligible ash aggregation effects) are found at all sampling sites, in good agreement with field observations (Poret et al., 2018). In addition, variations of the grain size distribution were accurately captured by the simulations as shown in Fig. 7b, which compares computed and observed particle distribution modes at all sites (S01-S10).

In terms of the normalised RMSE (see Sect. 4.3), the model results presented here show a marked improvement compared to the FALL3D-7.3 results reported by Poret et al. (2018). In fact, $RMSE_1$ and $RMSE_2$ reduce from 0.7 and 2.84 in Poret et al. (2018), to 0.58 and 0.98 in this study. The main differences between the simulations performed by Poret et al. (2018) and this study can be attributed to (i) the horizontal resolution of both the meteorological input data and the dispersal model; (ii) the vertical coordinate system; (iii) the TGSD used to define the source term; and (iv) the dry deposition parameterization. Specifically, higher resolution simulations along with the improved numerical scheme implemented in FALL3D-8.0 are expected to lead to a reduction of numerical diffusion errors, which is clearly manifested in the spatial distribution of the tephra mass loading field (compare Fig. 6 in the present study to Fig. 6 in Poret et al. (2018)).

### 5.4 Chernobyl-1986

Validation for the 1986 Chernobyl nuclear accident was carried out using the normalised RMSE with a constant absolute error (i.e. $RMSE_1$ with $w_1$). Since the measured radioactivity spans over four orders of magnitude, the $RMSE_1$ was calculated between the logarithms of the measured and observed radioactivity values at different tracking-points (29 for $^{131}$I, 44 for $^{134}$Cs,

and 45 for $^{137}$Cs, from the REM database). The obtained values of RMSE$_1$ are 0.10, 0.09 and 0.06 for $^{131}$I, $^{134}$Cs and $^{137}$Cs, respectively. The comparison of measured and simulated values for the best-fitted terminal velocity bins (Table 5) is shown in Fig. 8. Note that most of simulated values lie within an order of magnitude of the measurements (Fig. 9). Simulation results of the radioactive cloud evolution relative to $^{137}$Cs (vertically integrated radioactivity concentration in the atmosphere, expressed in $\mathrm{Bq\,m^{-2}}$) from 28 April to 9 May, 1986, are shown in Fig. 10. Figure 10 shows that the **Chernobyl-1986** simulations correctly reproduce the patterns described by Brandt et al. (2002). The evolution of the $^{137}$Cs dispersal is also available as a video in the Supplementary Material, together with videos corresponding to the dispersal of $^{134}$Cs and $^{131}$I.

## 6  Conclusions

Four different examples from the new FALL3D-8.0 benchmark suite have been presented to validate the accuracy of the latest major version release of the FALL3D model and complement a companion paper (Folch et al., 2020) on model physics and performance. In the first two examples, geostationary satellite observations from Meteosat-9 (SEVIRI) for the 2011 Puyehue-Cordón Caulle eruption (*i.e.* **Puyehue-2011** test suite case) and from Himawari-8 (AHI) for the 2019 Raikoke eruption (*i.e.* **Raikoke-2019** test suite case) were used to validate FALL3D simulations of far-range fine ash dispersal and SO$_2$ cloud dispersal, respectively. The metrics used for validation were the SAL and FMS and were applied to simulations with and without data insertion. To characterise the vertical structure of these volcanic clouds at selected data insertion times, geostationary satellite observations were collocated with CALIPSO overpasses. According to the SAL and FMS metrics, simulations initialised with data insertion consistently outperform simulations without data insertion. For the data insertion simulations, SAL remained below 1 out to 18 h (Table 4) and below 2 at lead times of 24 and 48 h for both ash and SO$_2$ simulations. While it is not yet clear what absolute values of SAL and FMS should represent an acceptable ash/SO$_2$ forecast, it is unlikely that in an operational setting a model simulation would be relied upon beyond 48 hours. Ideally, the model should be re-run with an updated data insertion time when new, good quality satellite retrievals become available. Based on our findings, it appears that SAL values of less than 1.5 and FMS values greater than 0.40 indicate good spatial agreement between the model and observation fields (*e.g.* Fig. 1b). However, it is important to consider that the satellite retrievals can be affected by several factors (*e.g.* cloud interference, high water vapour burdens, chosen detection thresholds) meaning that the ash/SO$_2$ detection schemes may miss some legitimate ash or SO$_2$ that the model is otherwise predicting (*e.g.* Fig. 1c). Limitations of the observations should also be taken into account when initialising simulations with data insertion. A data assimilation scheme that considers the errors in the satellite retrievals in addition to errors in the model simulations (*e.g.* using an ensemble approach to generate probabilistic forecasts) can be used to resolve these issues (e.g. Fu et al., 2017; Pardini et al., 2020).

For the **Etna-2013** test suite case, a very good agreement between field observations and simulations of deposit loads was found. This is evidenced by the fact that acceptable ratios of model to observed ash loading (between 1:3 and 3:1) exist across 4 orders of magnitude (*i.e.* from $10^{-3}$ $\mathrm{kg\,m^{-2}}$ to more than 10 $\mathrm{kg\,m^{-2}}$). Good agreement in terms of the local grain size distributions was also found for all sampling sites, as can be seen by comparing the simulated and observed mode of each distribution (Fig. 6b). Quantitative comparisons using normalised Root-Mean-Square Error (RMSE) metrics have shown that

the results of the present work have outperformed previous studies based on prior versions of FALL3D (version v7.3 or older). This is a direct consequence of multiple improvements incorporated in the new release of FALL3D (version v8.0), including a new numerical scheme and a new vertical coordinate system. Additionally, improvements in terms of code parallelism, memory management, model performance and scalability allow for higher resolution simulations and a more realistic modelling.

For the **Chernobyl-1986** test suite case, very good agreement between the model simulations and observations was found for the dispersal of radioactivity (*i.e.* $^{131}$I, $^{134}$Cs and $^{137}$Cs) resulting from the 22 April 1986 Chernobyl nuclear accident, consistent with the findings of Brandt et al. (2002).

Future developments of the test suite include adding more case studies, model inter-comparison studies that make use of the validation datasets provided here and validation of probabilistic forecasts. In terms of model utilities, we plan to introduce the option of ensemble forecasts and to incorporate data assimilation in future versions of FALL3D.

*Code and data availability.* FALL3D-8.0 and its test suite are available under the version 3 of the GNU General Public License (GPL) at https://gitlab.com/fall3d-distribution.

## Appendix A:  Volcanic ash retrieval

Satellite detection of volcanic ash using passive IR Brightness Temperature Differences (BTD) between channels centered around 11 and 12 $\mu$m has been widely used for more than 30 years (Prata, 1989a, b). Quantitative ash retrievals based on the BTD method have also been practiced for a long time (e.g. Wen and Rose, 1994; Prata and Grant, 2001) and the uncertainties stemming from detection (e.g. Simpson, 2000; Prata et al., 2001) and radiative transfer modelling (e.g. Wen and Rose, 1994; Corradini et al., 2008; Kylling et al., 2014; Stevenson et al., 2015; Western et al., 2015) are well-known. Figure A1 shows SEVIRI observations of the Cordón Caulle volcanic ash plume and illustrates the steps used to detect volcanic ash in the present study. For context, Fig. A1a and  A1b show a composite of MODIS true colour imagery and the SEVIRI 10.8 $\mu$m brightness temperature ($T_B^{11}$) respectively. Here, we propose an ash detection scheme based on applying successive masks that flag SEVIRI pixels as 'ash-affected' before attempting a subsequent quantitative ash retrieval:

1. First, we apply a temperature cut-off threshold to water vapor corrected BTDs ($\Delta T_{\text{ash}}$):

$$\Delta T_{\text{ash}} = T_B^{11} - T_B^{12} < T_{wc} \tag{A1}$$

that is, only those pixels with $\Delta T_{\text{ash}}$ less than the cut-off threshold of $T_{wc} = -0.5$ K are flagged as potential ash pixels. This water vapor correction follows the semi-empirical approach of Yu et al. (2002). As illustrated in Fig. A1c, this first threshold is reasonably effective at detecting the Cordón Caulle ash cloud. However, this simple cut-off threshold may not remove false positives due to temperature inversions generated by clear land at night (Platt and Prata, 1993), ice-covered surfaces (Yamanouchi et al., 1987), cold cloud-tops (Potts and Ebert, 1996) and high satellite zenith angles (Gu et al., 2005).

2. Second, we apply a cold surface mask designed to remove false positives due to reasons mentioned above. This cut-off condition relabels potential ash pixels as 'ash free' if:

$$\Delta T_{\text{ash}} > T_{sc} \text{ and } \begin{cases} T_B^{11} > 255 \text{ K over land} \\ T_B^{11} > 240 \text{ K over ocean} \end{cases} \tag{A2}$$

where $T_{sc} = -1.5$ K is the cold surface cut-off value. We note that $T_B^{11}$ condition of this mask will preserve ash detection sensitivity for high altitude (cold) ash clouds, which is particularly well-suited for the Cordón Caulle case study. However, this condition may not be suitable for low-level ash clouds (low thermal contrast resulting in less negative BTDs in addition to warmer cloud-tops). The effect of this mask is illustrated by comparing Figs. A1c and d. Note how, for the case shown, the cold surface mask removes almost all false positives over the region covered by low-level stratiform cloud.

3. Third, we apply a mask for false positives due to an increased path length at high satellite zenith angles (Gu et al., 2005). We mask out false positives at high zenith angles imposing:

$$\Delta T_{\text{ash}} > T_{zc} \text{ and } \zeta > 80^o \tag{A3}$$

where $T_{zc} = -2$ K is the zenith cut-off threshold and $\zeta$ is the satellite zenith angle. The effect of the high zenith mask can be seen by comparing Figs. A1d and e.

4. Finally, the last step in the detection process is to remove any spurious ash-labeled pixels using a noise filter that removes objects (groups of contiguous pixels) that are less than 16 pixels in size (Fig. A1f).

The MODIS true color composite shown in Fig. A1a illustrates that, even in a relatively complex scene (numerous clouds, large regions of land and ocean, high mountains, ice-covered surfaces, etc.), the ash detection is robust and provides a good balance between reduced false positives and increased true positives. An interesting point to note is that negative BTDs in the vicinity of Cordón Caulle are enhanced due to the high satellite zenith angles at these locations. Gu et al. (2005) discuss the benefit of improved sensitivity to ash at high satellite zenith angles, but also show that mass loading retrievals can be overestimated in these situations. We correct for the effect of high zenith angles after retrieving the mass loading. Once pixels have been identified as being 'ash-affected' we apply a Look-up Table (LuT) approach (Prata and Grant, 2001; Prata and Prata, 2012) to retrieve volcanic ash optical depth ($\tau$), effective radius ($r_e$; in µm), and column mass loading ($m_l$; in $\text{g m}^{-2}$). The retrieval procedure is illustrated in Fig. A2a. The temperature difference model employed here is based on the forward model developed by Prata (1989b) and Wen and Rose (1994):

$$I_\lambda \approx e^{-\tau(\lambda)} B(T_s) + \left(1 - e^{-\tau(\lambda)}\right) B(T_c) \tag{A4}$$

where $I_\lambda$ is the radiance at the top of the atmosphere at wavelength ($\lambda$), $\tau_\lambda$ is the wavelength-dependent optical depth, $B(T_s)$ is the Planck radiance evaluated for surface temperature ($T_s$) below the ash cloud, and $B(T_c)$ is the Planck radiance corresponding

to the temperature at the ash cloud-top ($T_c$). The optical depth is defined as:

$$\tau(\lambda) = \pi L \int_0^\infty r^2 Q_{ext}(\lambda, r) n(r) dr \tag{A5}$$

where $L$ is the geometric thickness of the ash cloud, $Q_{ext}(\lambda, r)$ is the extinction efficiency factor (determined from Mie calculations), $r$ is the particle radius and $n(r)$ represents the distribution of particles within the ash cloud. The ash mass loading is determined as:

$$m_l = \frac{4}{3} \rho \frac{r_e \tau(\lambda)}{Q_{ext}(\lambda, r_e)} \cos(\zeta), \tag{A6}$$

where $\rho$ is the ash particle density (set to $2500 \ \mathrm{kg \, m^{-3}}$ based on field measurements reported by Dominguez et al. (2020) for distal ash) and the $\cos(\zeta)$ term corrects the mass loading based on the satellite zenith angle. Uncertainties using this approach have been previously estimated to be up to 50% (Wen and Rose, 1994; Corradini et al., 2008). Our microphysical model, used to parameterize a volcanic ash cloud in the radiative transfer calculations, assumes that ash particles are spherical, composed of andesite and conform to a lognormal size distribution with a spread equal to 0.5 (geometric standard deviation $\sigma_g = 1.65$), similar to existing operational volcanic ash retrieval algorithms (e.g. Francis et al., 2012; Pavolonis et al., 2013).

The retrieval scheme relies on interpolating pre-computed LuTs generated by conducting radiative transfer calculations made for varying values of $r_e$, $\tau$, $T_s$ and $T_c$. The LuTs are generated using the approach of Prata (1989b) to solve the radiative transfer equation for a single-layer ash cloud using the Discrete Ordinates Method (DOM; Stamnes et al., 1988; Laszlo et al., 2016). In the present study, we consider $\tau$ in the range from 0–9.9 in steps of 0.1 and $r_e$ from 1–15 $\mu$m in steps of 0.2 $\mu$m. All radiative transfer calculations use 16 radiation streams and a unique LuT is generated for every combination of $T_c$ and $T_s$ identified from ash-affected pixels. Figure A3 shows a graphical example of a LuT generated for one combination of $T_s$ and $T_c$ and the range of $\tau$ and $r_e$ considered.

To determine $T_s$ directly from measurements it is generally recommended to find a clear-air pixel near the volcanic cloud of interest (e.g. Wen and Rose, 1994) and can sometimes be determined by finding the maximum value of $T_B^{11}$ in the scene (Prata and Lynch, 2019). Obtaining an estimate for $T_c$ from measurements, however, can be more difficult as the minimum value of $T_B^{11}$ may not correspond to the (semi-transparent) ash cloud of interest. Nevertheless, even if $T_s$ and $T_c$ can be reasonably estimated from measurements, it is often assumed that a single or mean value (and corresponding standard deviation) is representative of the entire ash cloud. Figure A1 shows that, in our case, the ash plume extends more than 60° in longitude and 20° in latitude, over land (including the Andes mountain ranges) and ocean, meaning that there is a considerable variation in cloud-top and surface temperature across ash-affected pixels. In addition, the meteorological setting within the considered spatial and temporal domains is complex (significant amounts of clouds), making estimates of $T_s$ and $T_c$ from measurements challenging for the Cordón Caulle case study. To account for variation in $T_s$ and $T_c$ across space and time, we use ERA5 reanalysis data Copernicus Climate Change Service (C3S) (2017) to determine $T_s$ and $T_c$ at every ash-affected pixel over our study period from 5–10 June 2011 (in one hour time steps).

To determine $T_s$ from ERA5, we use the surface skin temperature ($T_{skin}$) and assume that the atmospheric transmittance ($t_{atm}$)

has only a small effect on measured radiances at the top of the atmosphere for split-window channels (*i.e.* $t_{atm} \approx 1$). We also correct $T_{skin}$ for variations in land surface emissivity using the University of Wisconsin global IR land surface emissivity database (Seemann et al., 2008). For ocean surfaces, we set the emissivity to 0.99 consistent with Western et al. (2015). Analysis comparing $T_B^{11}$ SEVIRI measurements against the emissivity-corrected $T_{skin}$ for clear-sky pixels on 4 June 2011 indicates average differences of $\sim$2 K. To determine $T_c$ from ERA5 we require an estimate of the volcanic cloud-top height. A fortuitous CALIPSO pass early on during the eruption on 5 June 2011 reveals that the Cordón Caulle ash cloud reached as high as 13–14 km above sea level (Fig. A4a) and later observations indicate heights from 10–13 km (Fig. A4b). For the retrievals presented here, we take $T_c$ to be the ERA5 temperature at 13 km (a. s. l.) and make the simplifying assumption of constant height at all locations (and times) for every ash-affected pixel detected during 5–10 June 2011. The assumption of constant cloud-top height allows $T_c$ to vary in time, horizontally but not vertically. However, FALL3D simulations indicate that the height of the Cordón Caulle ash cloud was relatively stable over the course of its dispersion in the atmosphere and so we expect errors introduced by this assumption to be small. This was probably due to its injection into the stratosphere and its transport via the Southern hemisphere jet stream (height variations from 11–15 km; Klüser et al., 2013; Vernier et al., 2013; Prata et al., 2017). For our study period from 5–10 June 2011, $T_c$ ranged from 206–226 K while $T_s$ ranged from 230–304 K. We therefore performed radiative transfer calculations to construct unique LuTs, in steps of 2 K, for every possible combination of $T_s$ and $T_c$ within these ranges.

## Appendix B: SO₂ retrieval

To retrieve total SO₂ column densities (in Dobson Units; DU), we use the three-channel technique of Prata et al. (2003). This retrieval scheme exploits the strong SO₂ absorption feature near 7.3 μm and is only sensitive to upper-level SO₂ ($> 4$ km) due to the absorption of lower-level water vapor at this wavelength. The three channels used to detect SO₂ using AHI measurements are centred around 6.9, 7.3 and 11.2 μm. To determine whether there is an SO₂ signal in the data, we first construct a synthetic 7.3 μm brightness temperature by interpolating from 6.9 to 11.2 μm in the radiance space and then converting to brightness temperature via the Planck function (Prata et al., 2003). Figure B1 illustrates how the interpolation procedure works in radiance space. The resulting 'clear' brightness temperature ($T_{BC}^{7.3}$) is a good approximation of the measured value ($T_B^{7.3}$) in a SO₂-free atmosphere, so that one can identify SO₂ clouds by taking the difference between these two variables:

$$\Delta T_{\text{SO}_2} = T_{BC}^{7.3} - T_B^{7.3} \tag{B1}$$

In theory, $\Delta T_{\text{SO}_2}$ should be equal to zero under clear-sky conditions and increase with increasing SO₂ column density. However, in reality, high satellite zenith angles and variations in temperature and humidity can cause $\Delta T_{\text{SO}_2}$ to be positive even under clear-sky conditions (Prata et al., 2003; Doutriaux-Boucher and Dubuisson, 2009). To remove false positives due to high satellite zenith angles and high water vapour burdens, we compute two SO₂-related BTDs ($\Delta T_{69}$ and $\Delta T_{86}$) and apply two successive temperature cut-off thresholds:

$$\Delta T_{69} = T_B^{6.9} - T_B^{7.3} > T_{69} \tag{B2}$$

where only those pixels with a $\Delta T_{69}$ greater than a cut-off threshold of $T_{69} = -2.5$ K are flagged as potential $SO_2$. The second threshold takes advantage of the $SO_2$ absorption feature near 8.6 µm:

$$\Delta T_{86} = T_B^{11} - T_B^{8.6} > T_{86} \tag{B3}$$

where only those pixels with a $\Delta T_{86}$ greater than a cut-off threshold of $T_{86} = 3.5$ K are flagged as potential $SO_2$. In addition, the presence of water/ice clouds and embedded volcanic ash can also affect the interpolation procedure used to construct $T_{BC}^{7.3}$. Figures B2a to B2c show, respectively, $T_B^{7.3}$, $T_{BC}^{7.3}$, and $\Delta T_{SO_2}$ brightness temperatures for the $SO_2$-rich Raikoke cloud on 22 June 2019 at 21:00 UTC. Clearly, the interpolation procedure does a good job at removing the $SO_2$ signal from the measurements resulting in excellent detection sensitively for $\Delta T_{SO_2}$. Comparison of Figures B2d, e and f show how the $\Delta T_{69}$ and $\Delta T_{86}$ thresholds are used to remove false alarms whilst preserving legitimate $SO_2$-affected pixels. Considering that $\Delta T_{SO_2}$ calculated via Eq. (B1) is a function of the total column density of $SO_2$, we can retrieve the total column amount by constructing this function from offline radiative transfer calculations. For this purpose we use the MODTRAN-6.0 code (Berk et al., 2014) to compute top-of-the-atmosphere (TOA) radiances at the 7.3 µm wavelength (Fig. A2b). All radiances determined from MODTRAN-6.0 were convolved using the AHI spectral response functions. These radiances are then converted to brightness temperatures to compute BTDs between an $SO_2$-free atmosphere and atmospheres with varying column densities of $SO_2$ at 7.3 µm (i.e. $\Delta T_{SO_2}$). We are then able to generate a function representing the relationship between the $SO_2$ column density, $u(\Delta T_{SO_2})$, and $\Delta T_{SO_2}$ by interpolating between the data points generated from the radiative transfer modelling (Fig. B4). In practice we generate this function using a 1D quadratic interpolation procedure implemented in the SciPy python package (Virtanen et al., 2020). Atmospheric profiles of temperature, humidity and gases ($H_2O$, $CO_2$, $O_3$, $N_2O$, CO and $CH_4$) were taken from the US standard atmosphere. In varying the total $SO_2$ column densities (in DU), we must specify an $SO_2$ profile. We use CALIPSO total attenuated backscatter profiles collocated with Himawari-8 observations of $\Delta T_{SO_2}$ to constrain the $SO_2$ profiles used in the radiative transfer calculations for the Raikoke case. Figure B3a shows a daytime CALIOP overpass intersecting $SO_2$ layers detected by Himawari-8 during the initial explosive phase of the Raikoke eruption. The vertical distribution of cloud/aerosol layers in the CALIOP observations reveals that the eastern part of the plume reached at least 12 km (a.s.l.). Later CALIOP/AHI observations reveal complex stratospheric dynamics; two distinct components are apparent in the attenuated backscatter data with thin layers (1–2 km) present at 13–15 km in the northern part of the $SO_2$ cloud and ~12 km in the southern part (Fig. B3b). Based on these initial observations, we constructed $u(\Delta T_{SO_2})$ using a uniform $SO_2$ distribution with a maximum cloud-top height of 13.5 km and thickness of 2.5 km (Fig. B4). The retrieval then proceeds by computing $\Delta T_{SO_2}$ from AHI data and evaluating $u(\Delta T_{SO_2})$ for every $SO_2$-affected pixel.

*Author contributions.* AF and AC conceived the study and planned the test cases. AF and LM wrote the bulk of FALL3D-8.0 code, with contributions of GM. ATP conducted the satellite retrievals and implemented the validation metrics. LM, AF and GM ran model executions and validations. AC and GM ran the simulations for radionuclide dispersion. All authors have contributed to the writing of the text.

*Competing interests.* The authors declare no competing interests.

*Acknowledgements.* This work has been partially funded by the H2020 Center of Excellence for Exascale in Solid Earth (ChEESE) under
595 the Grant Agreement 823844. ATP acknowledges funding from the European Union's Horizon 2020 research and innovation programme
under the Marie Skłodowska-Curie grant agreement H2020-MSCA-COFUND-2016-754433. AC and GM acknowledge the European project
EUROVOLC (grant agreement number 731070) and the Ministero dell'Istruzione, dell'Università e della ricerca (MIUR, Roma, Italy)
Ash-RESILIENCE project (grant agreement number 805 FOE 2015). We acknowledge the use of the ERA5 Fifth generation of ECMWF
atmospheric data from the Copernicus Climate Change Service; neither the European Commission nor ECMWF is responsible for the use
made of the Copernicus Information and Data. The European Space Agency (ESA) and EUMETSAT are thanked for supplying the SEVIRI
satellite data and the Japan Aerospace Exploration Agency (JAXA) and the Japanese Meteorological Agency (JMA) are thanked for supplying
the Himawari-8 data used in this study.

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

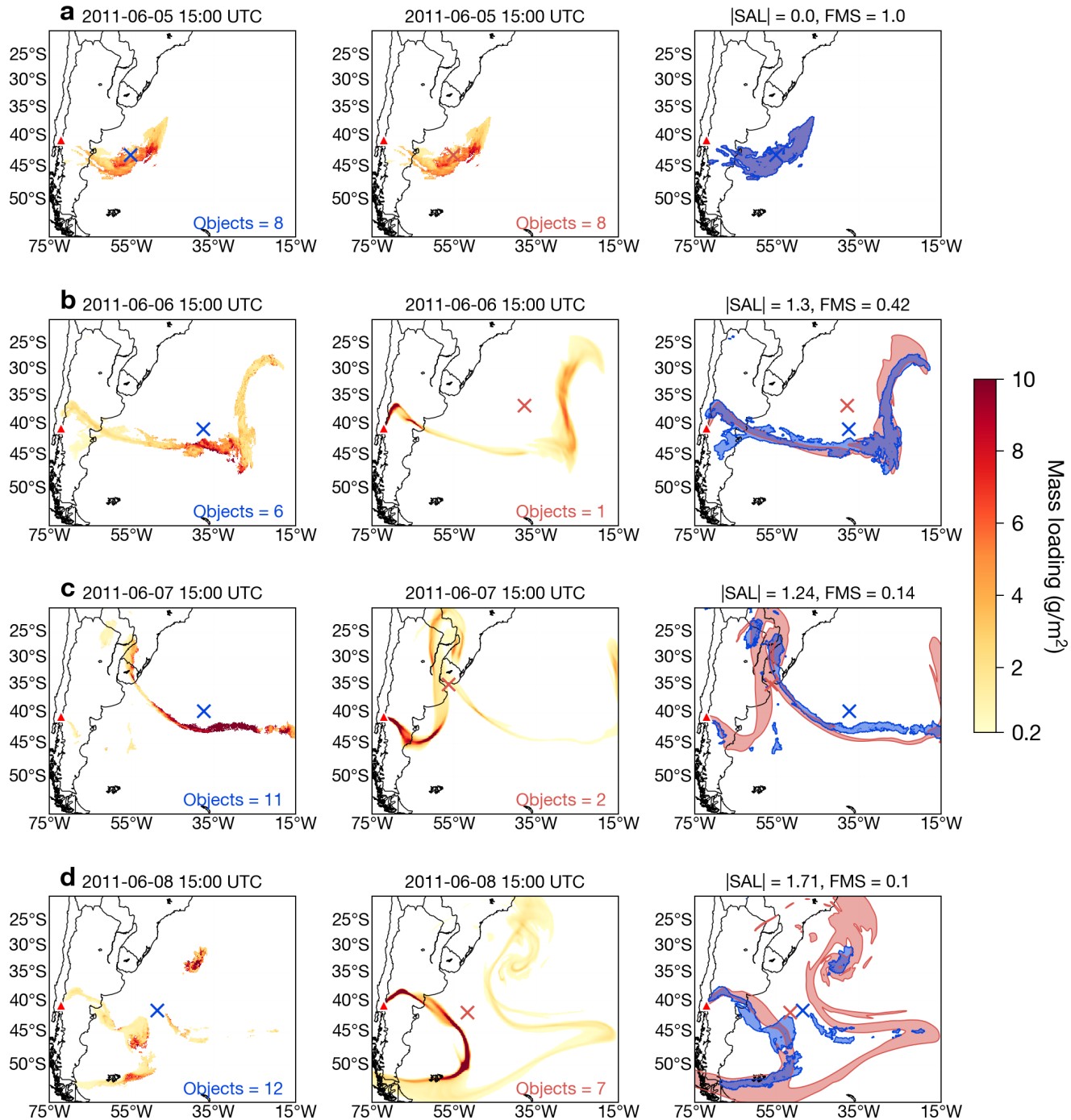

**Figure 1. Puyehue-2011** test validation results for fine ash mass loading using SEVIRI retrievals on (a) 5 June 2011 at 15:00 UTC (data insertion time), (b) 6 June 2011 at 15:00 UTC, (c) 7 June 2011 at 15:00 UTC and (d) 8 June 2011 at 15:00 UTC. Left panels show satellite retrievals with 0.2 g m$^{-2}$ contour in blue and centre of mass indicated with 'x'. Middle panels show FALL3D fine ash mass loading model simulation (0.2 g m$^{-2}$ contour in red). Right panels show spatial overlap of model (shaded in red) vs. observed (shaded in blue) fields with validation metrics annotated (see Sect. 4 for details). A full animation of the data insertion simulations is available in the Supplementary Material.

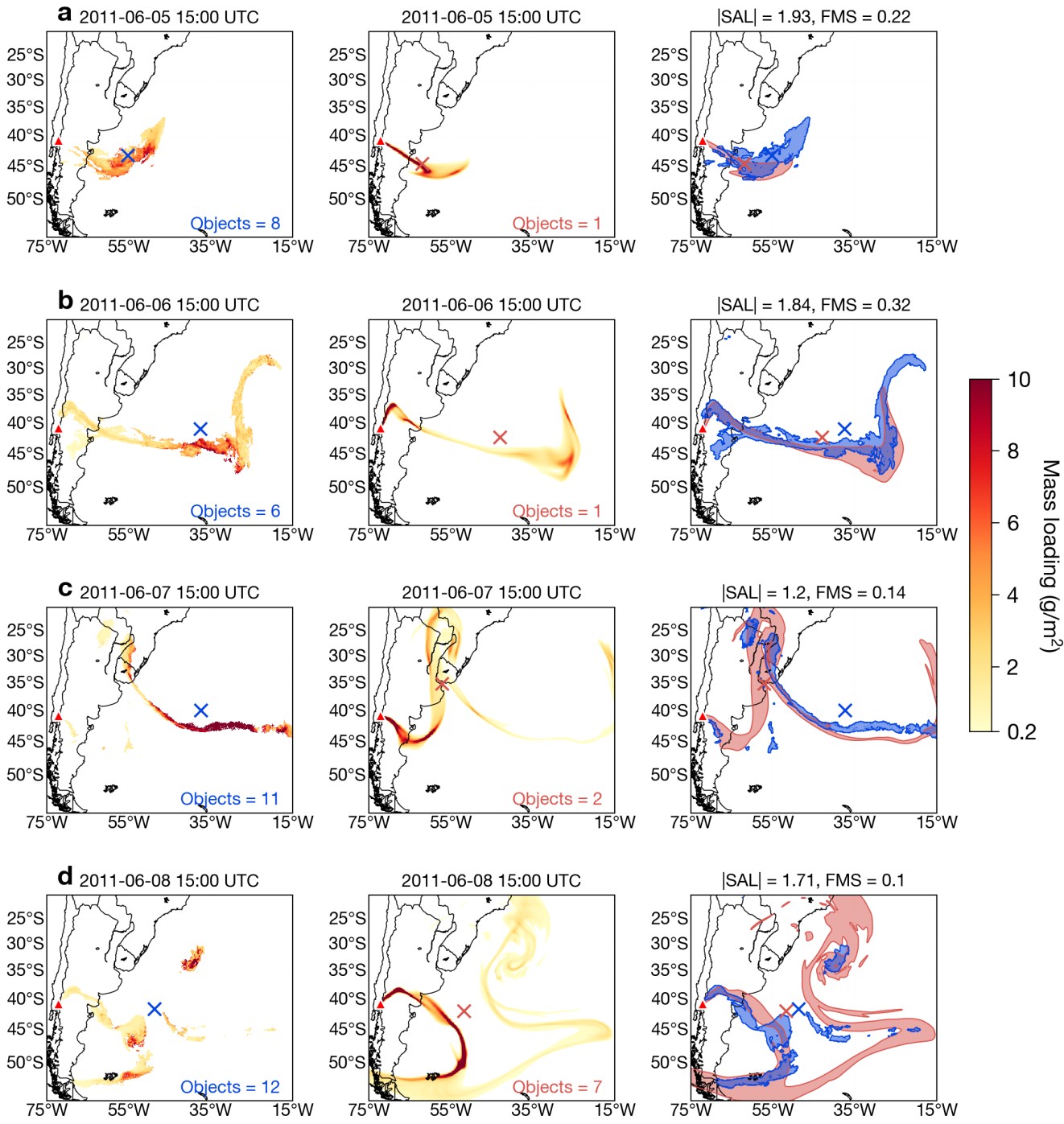

**Figure 2.** Same as Fig. 1 but without data insertion.

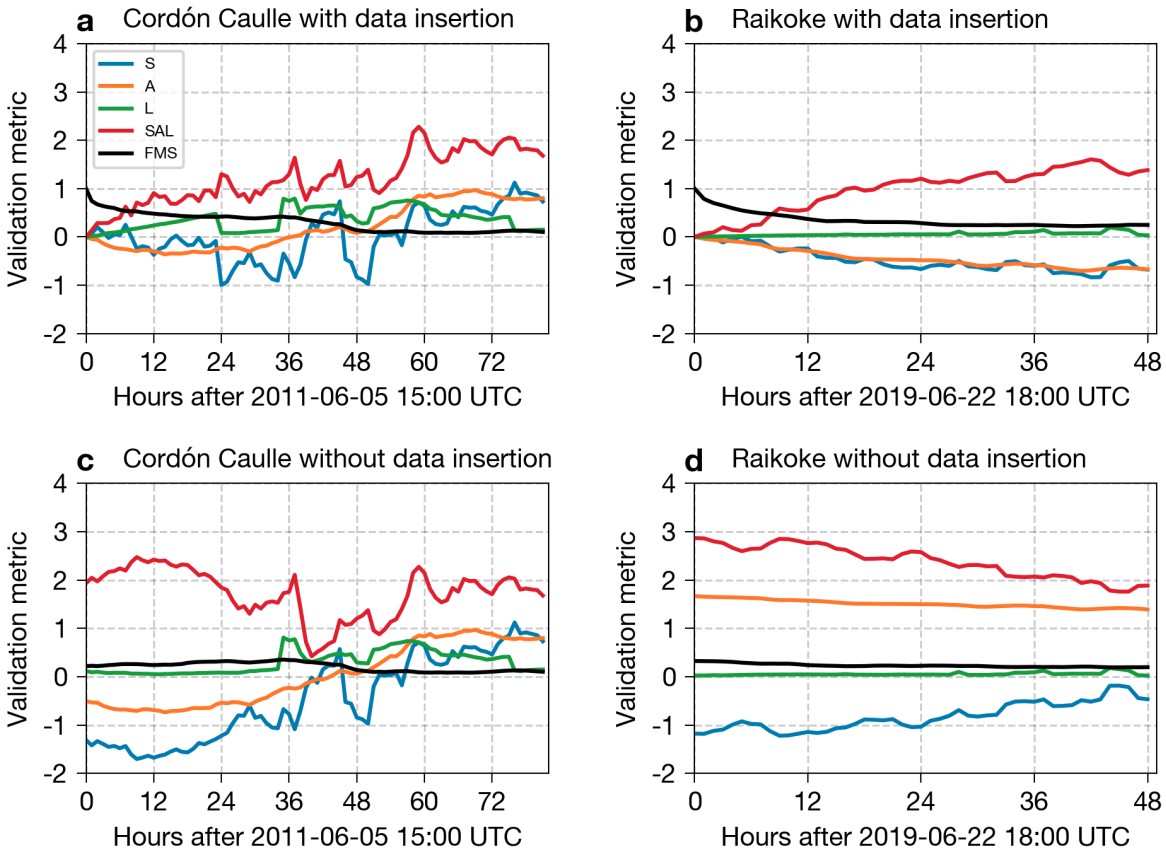

**Figure 3.** Time-series of validation metrics for (a) **Puyehue-2011** test case with data insertion, (b) **Raikoke-2019** test case with data insertion, (c) **Puyehue-2011** without data insertion and (d) **Raikoke-2019** without data insertion.

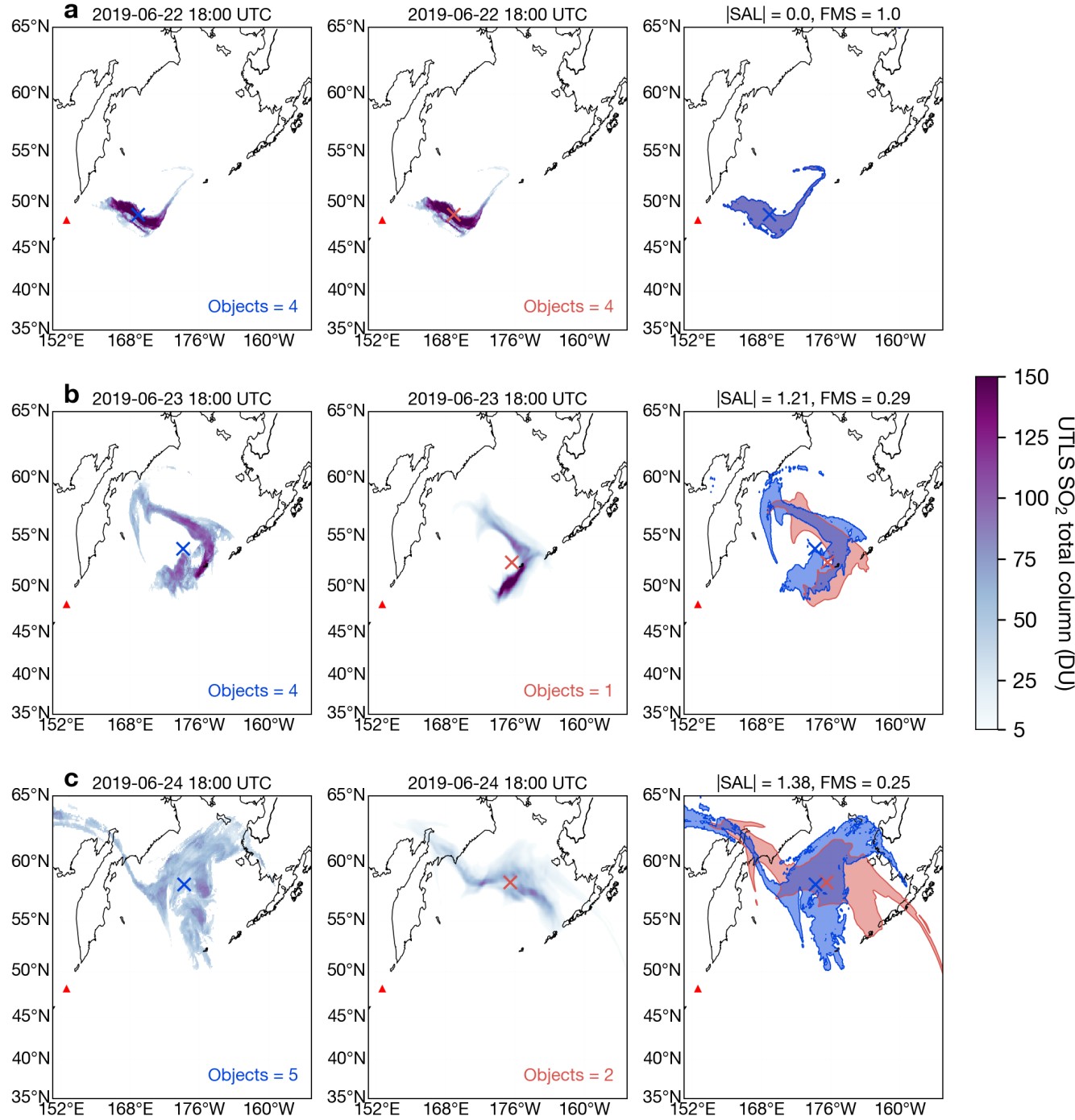

**Figure 4. Raikoke-2019** validation test results using AHI upper-troposphere lower-stratosphere (UTLS) total column burdens retrievals (DU) on (a) 22 June 2019 at 18:00 UTC (data insertion time), (b) 23 June 2019 at 18:00 UTC and (c) 24 June 2019 at 18:00 UTC. Left column shows satellite retrievals with 5 DU contour in blue and centre of mass indicated with 'x'. Middle column shows FALL3D model simulation (5 DU contour in red). Right column shows spatial overlap of model (shaded in red) vs. observed (shaded in blue) fields. A full animation of the data insertion simulations is available in the Supplementary Material.

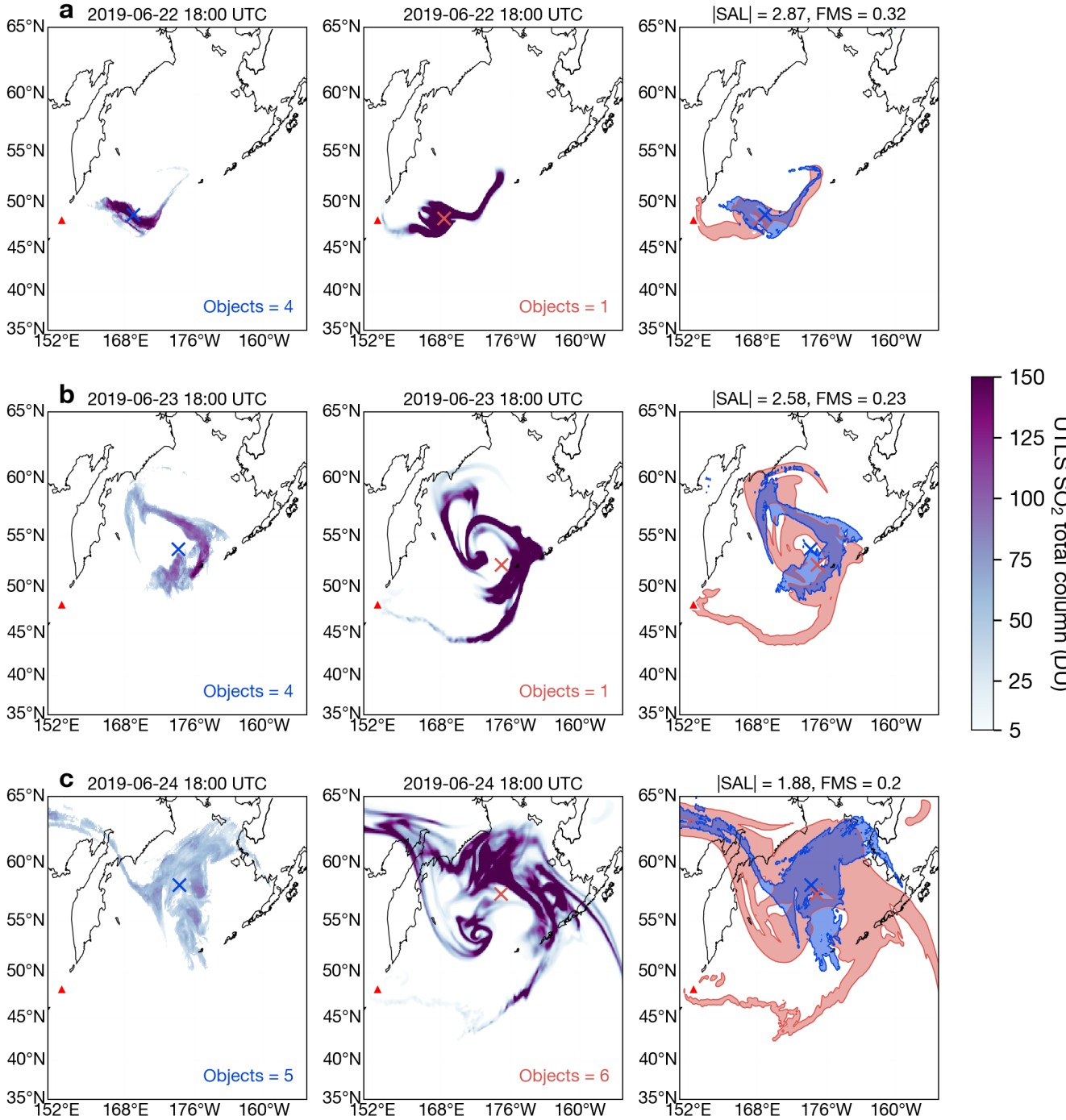

**Figure 5.** Same as Fig. 4 but without data insertion.

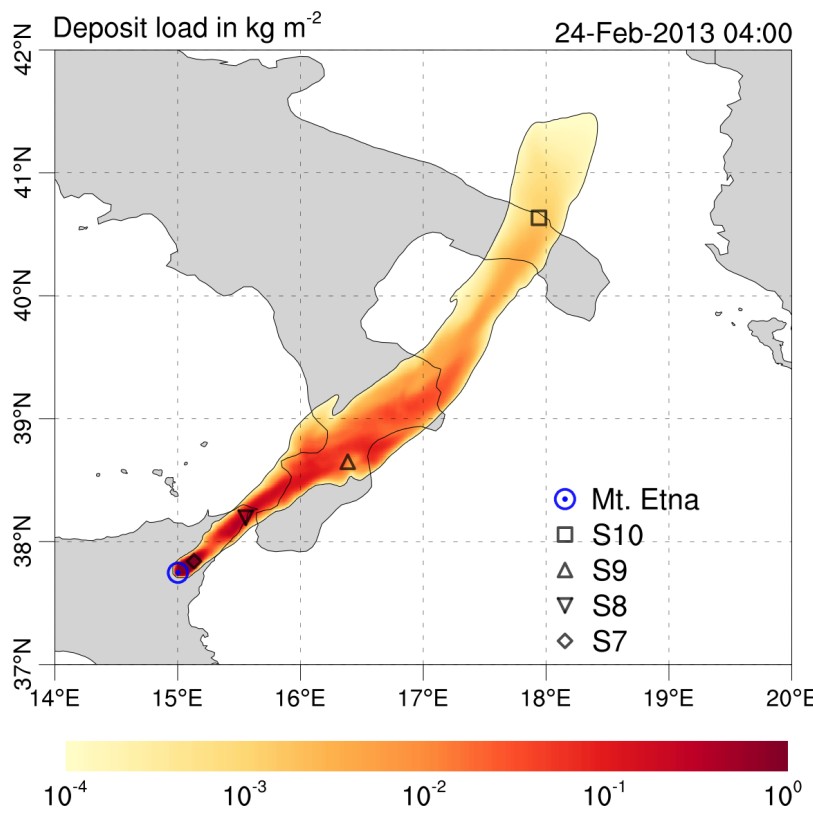

**Figure 6.** Tephra fallout deposit from test case **Etna-2013**. The spatial distribution of the modelled tephra loading coincides with the locations of sampling sites. Sites S7-S10 are indicated in the map by symbols. Proximal sites S1-S6 ($<$ 16km from the source) not shown for clarity.

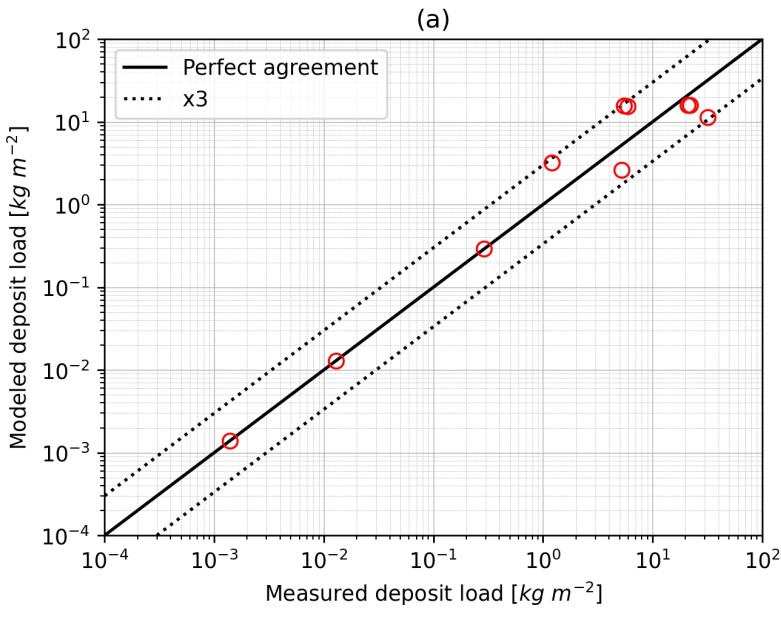

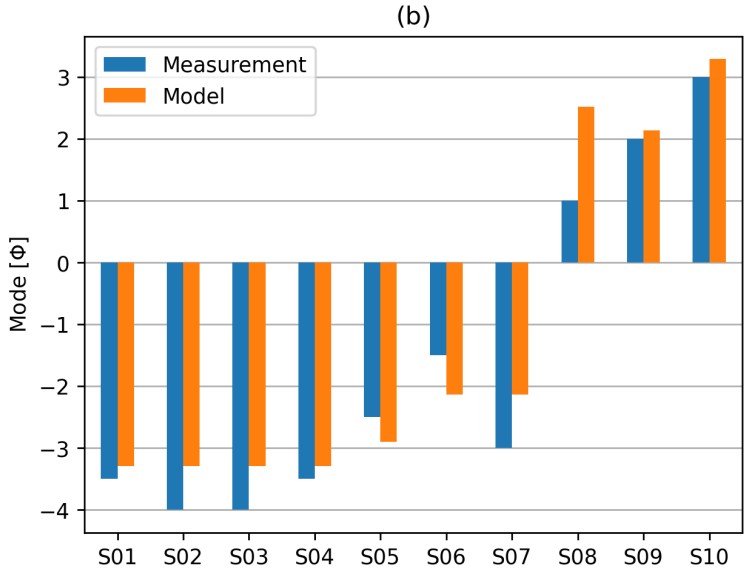

**Figure 7.** Comparison between field data at 10 sampling sites (S1-S10) and results from **Etna-2013** test case. (a) Tephra mass loading. (b) Mode of the distributions. Field data were obtained from the sample dataset reported by Poret et al. (2018).

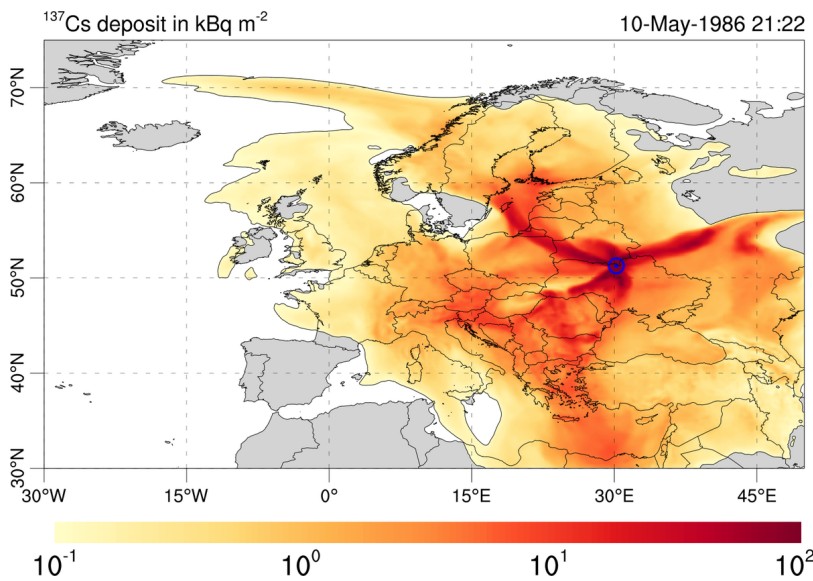

**Figure 8. Chernobyl-1986** accumulated total deposition of $^{137}$Cs on 10 May 1986. The underlying map is reported just for reference and could contain nations borders that are under dispute.

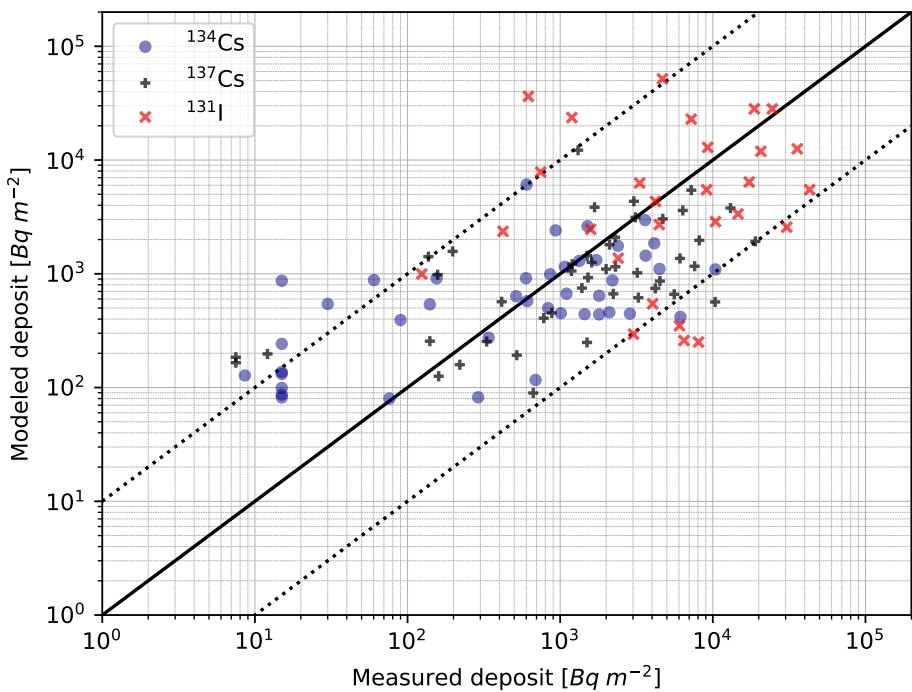

**Figure 9.** Comparison between measurements and **Chernobyl-1986** test deposit results at different locations of $^{137}$Cs, $^{134}$Cs and $^{131}$I on 10 May 1986. Solid line represents one-to-one agreement between measured and observed values. Lower and upper dashed lines represent a factor of 0.1 and 10, respectively, from the one-to-one line.

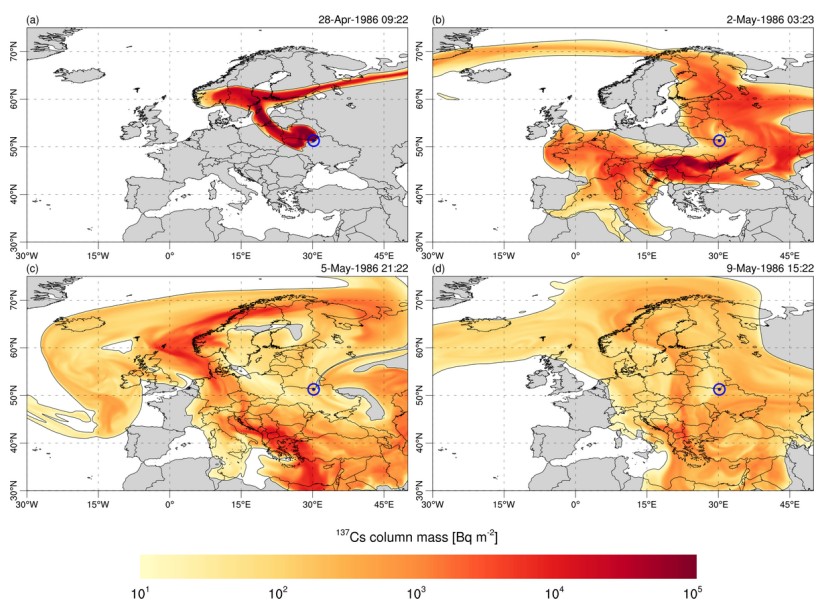

**Figure 10.** Vertically integrated $^{137}$Cs radioactivity concentration in the atmosphere at different times according to **Chernobyl-1986** test results. (a) 28 April 1986, (b) 2 May 1986, (c) 5 May 1986 and (d) 9 May 1986. The underlying maps are reported just for reference and could contain nations borders that are under dispute.

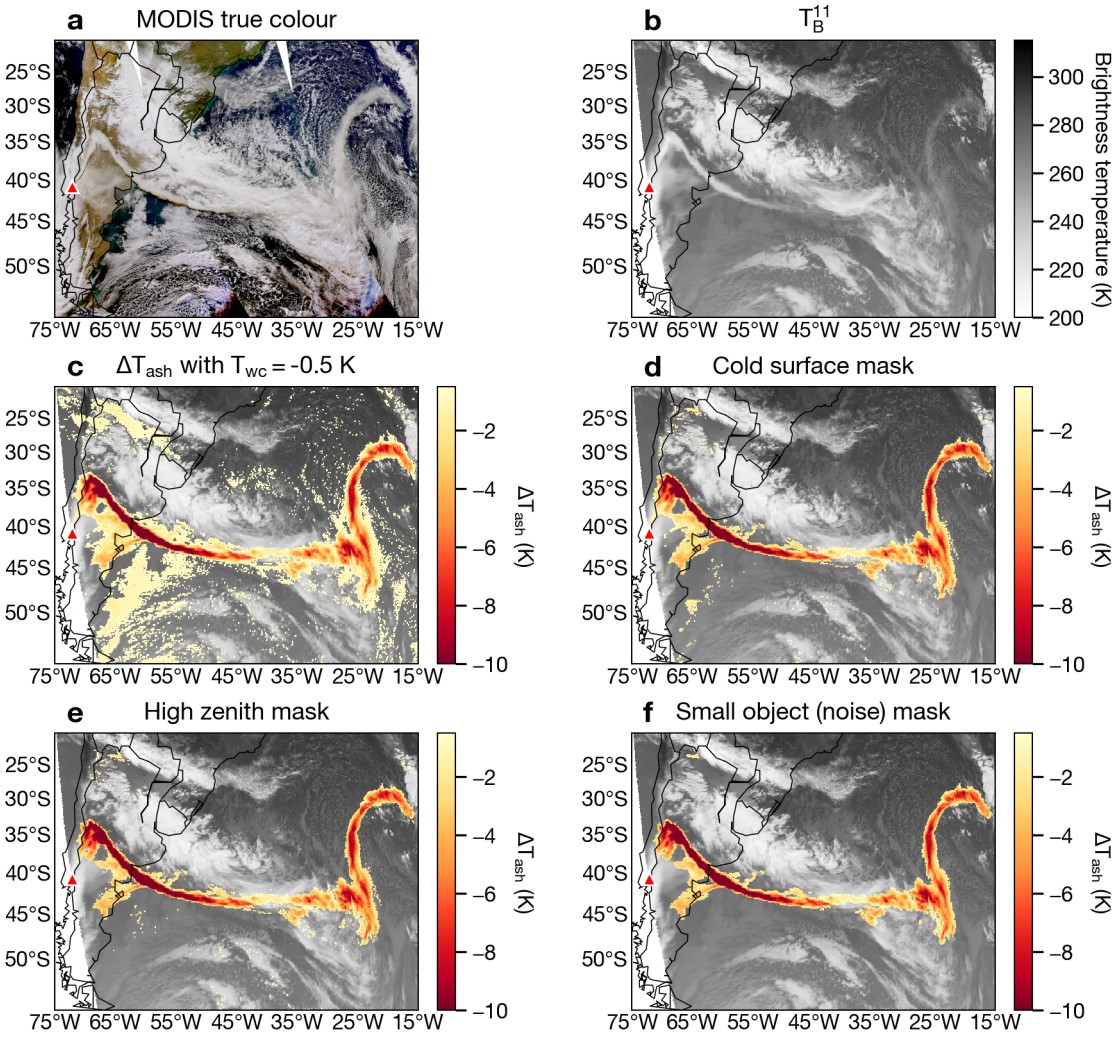

**Figure A1.** Volcanic ash detection scheme for the Puyehue-Cordón Caulle (indicated by the triangle on each map) eruption. (a) MODIS true color composite from 2011-06-06 at 15:15–18:40 UTC. (b) SEVIRI 10.8 $\mu$m brightness temperature ($T_B^{11}$) at 2011-06-06 18:45 UTC. (c) Same as (b) with water-vapor corrected BTD ($\Delta T_{ash} = T_B^{11} - T_B^{12}$) overlaid. (d), (e) and (f) are the same as (c) with cold surface, high zenith and noise masks applied, respectively.

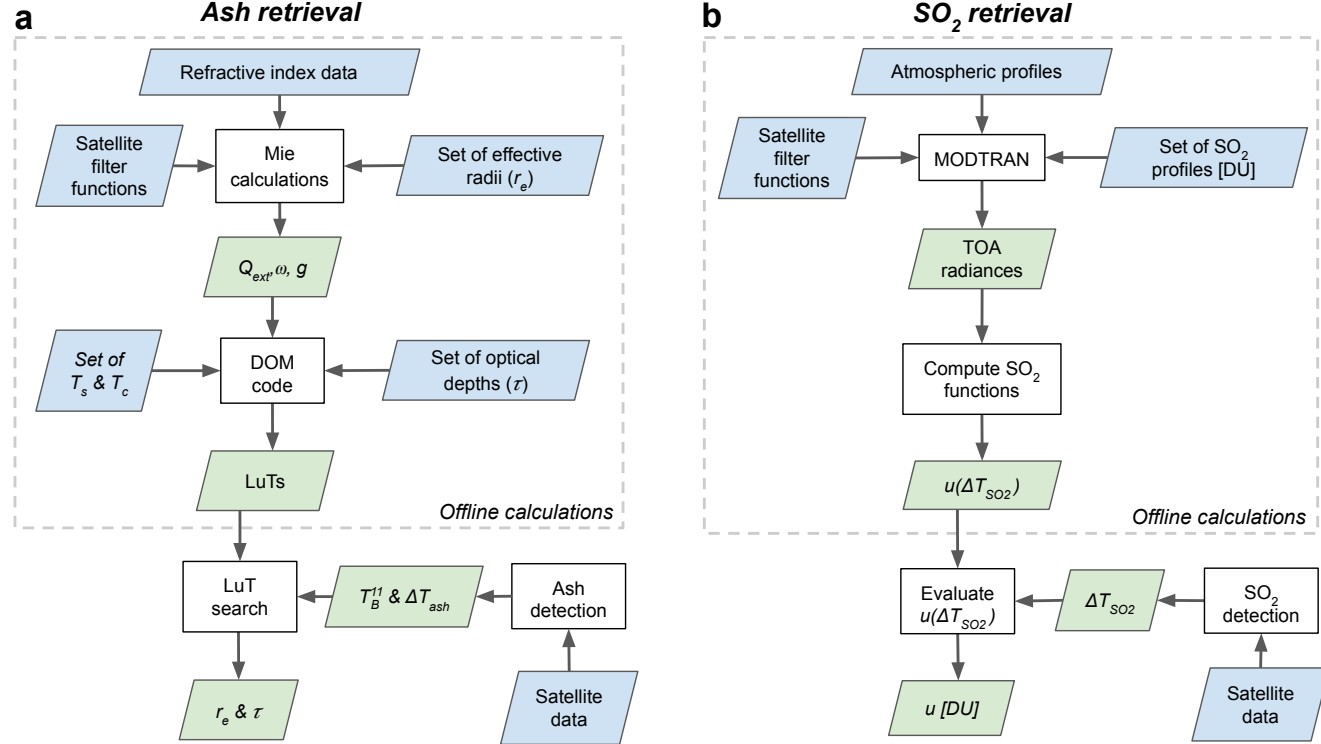

**Figure A2.** Flow diagram showing the (a) volcanic ash retrieval process and (b) volcanic SO$_2$ retrieval process used in the present study. Parallelograms indicate datasets (blue for inputs, green for outputs) and rectangles indicate processes (i.e. code used to implement the retrieval algorithms and perform radiative transfer calculations). Offline calculations are any computations that are pre-computed (i.e. before any observations are made by the satellite).

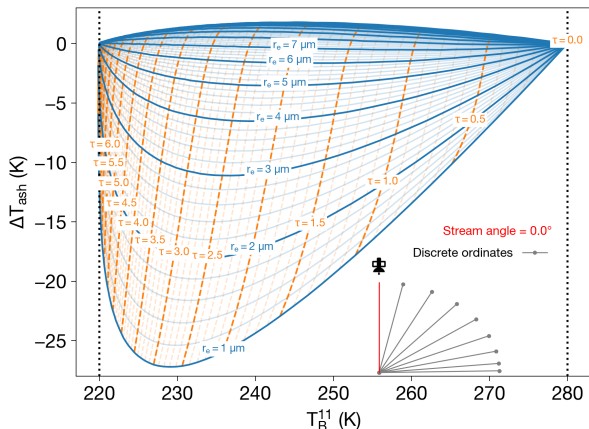

**Figure A3.** Graphical illustration of a volcanic ash look-up table for a surface temperature $T_s = 280$ K and cloud-top temperature, $T_c = 220$ K. Dashed near-vertical lines indicate lines of constant optical depth, $\tau$, and solid U-shaped curves indicate lines of constant effective radius, $r_e$.

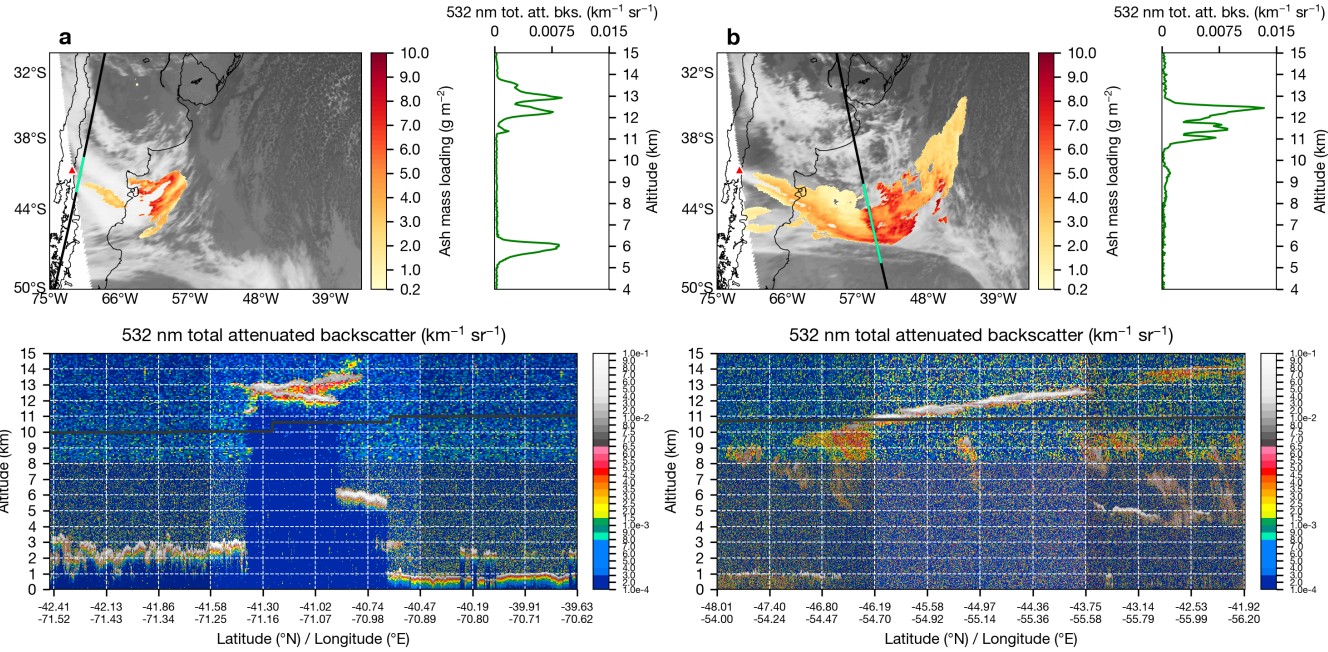

**Figure A4.** (a) SEVIRI ash mass loadings and CALIOP vertical profile of the Cordón Caulle ash plume on 5 June 2011 at 06:00 UTC. Top left panel: Mass loading retrievals (yellow-orange-red color scale) with brightness temperatures plotted underneath (red triangle indicates location of Cordón Caulle). Black line indicates CALIOP track and green highlight indicates full latitude/longitude range displayed on the bottom panel. Top right panel: 532 nm total attenuated backscatter profile averaged over the latitude/longitude range highlighted on bottom panel. Bottom panel: 532 nm total attenuated backscatter curtain (black line indicates tropopause). (b) Same as (a), but for 5 June 2011 at 18:00 UTC.

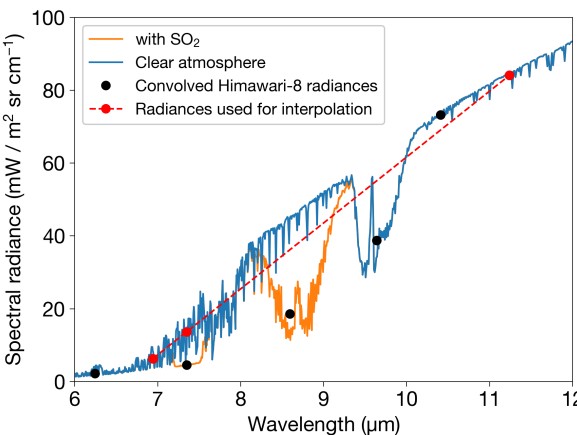

**Figure B1.** MODTRAN6.0 simulations for atmospheres with and without $SO_2$ demonstrating how the interpolation procedure is used to estimate a clear-sky radiances from an atmosphere with $SO_2$. Convolved radiances were derived using the Himawari-8 spectral response functions.

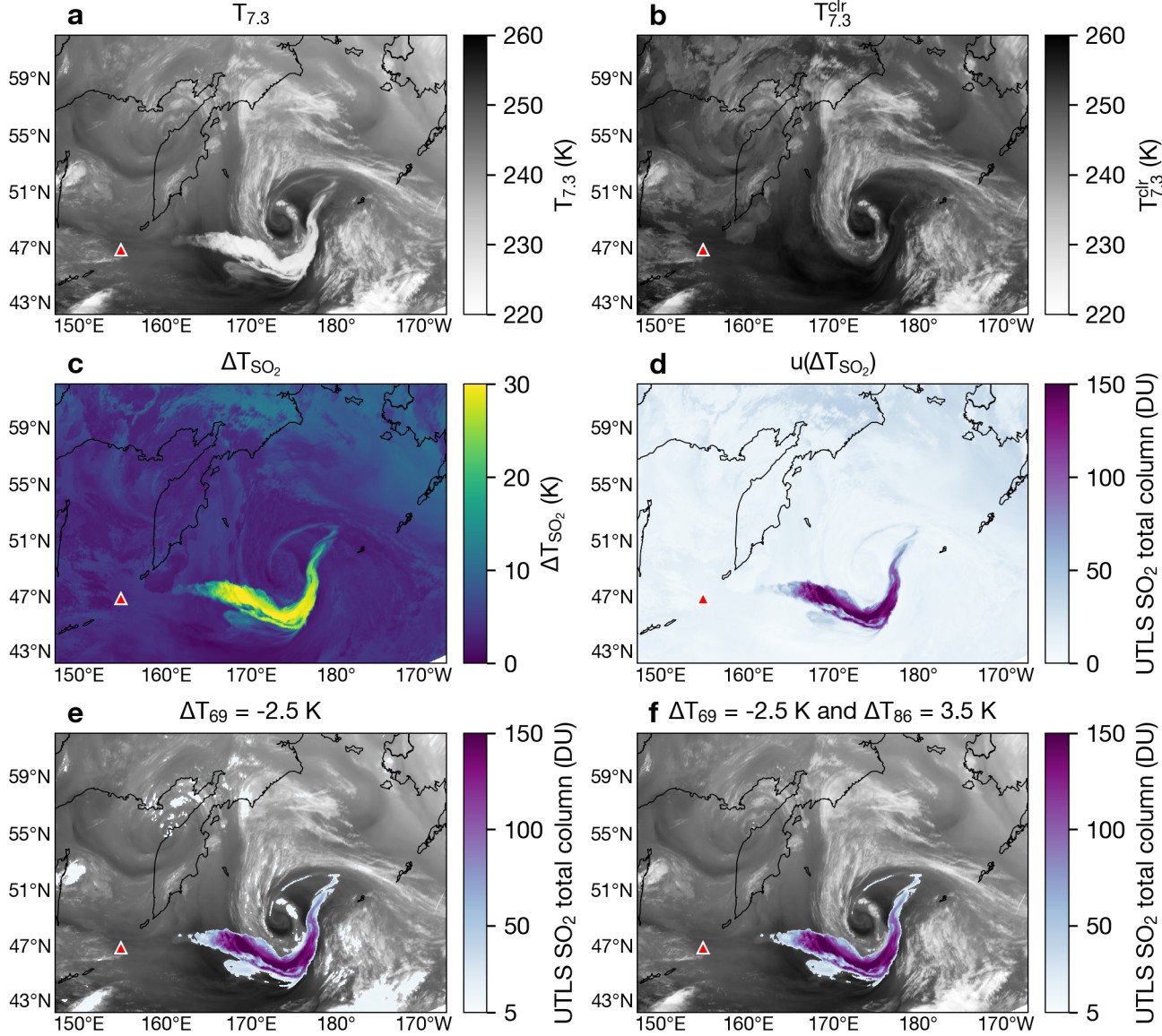

**Figure B2.** Volcanic SO₂ detection scheme applied to Himawari-8 observations of the Raikoke SO₂ cloud on 22 June 2019 at 21:00 UTC. Location of Raikoke volcano indicated by red triangle on each map. (a) Himawari-8 brightness temperature at 7.3 $\mu$m ($T_B^{7.3}$). (b) Synthetic 7.3 $\mu$m brightness temperature ($T_{BC}^{7.3}$) determined from interpolation procedure (see Sect. B for details). (c) SO₂ BTD ($\Delta T_{SO_2} = T_B^{7.3} - T_{BC}^{7.3}$). (d) SO₂ total column loading, $u(\Delta SO_2)$, (see Sect. B for details). (e) $u(\Delta SO_2)$ with $\Delta T_{wv} = -2.5$ K and 5 DU thresholds. (f) $u(\Delta SO_2)$ with $\Delta T_{wv} = -2.5$ K, $\Delta T_{86} = 3.5$ K and 5 DU thresholds.

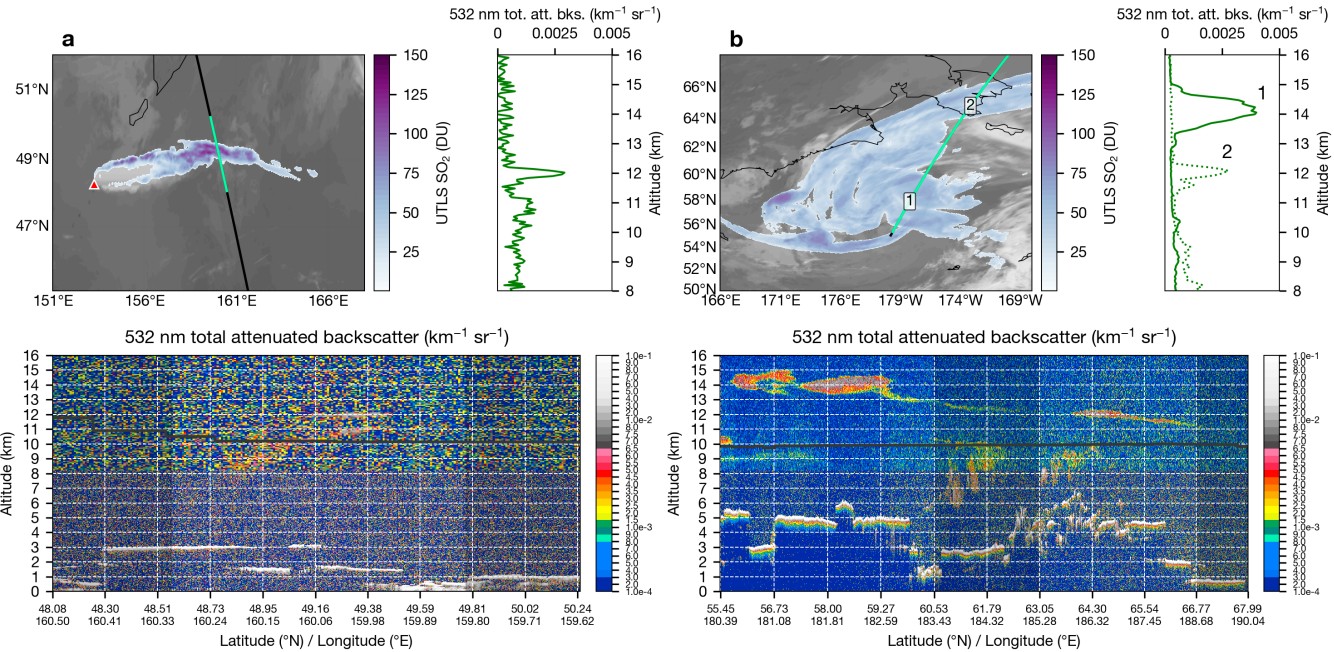

**Figure B3.** (a) AHI $SO_2$ upper-troposphere lower-stratosphere (UTLS) total column burdens (DU) and CALIOP vertical profile of the Raikoke $SO_2$ plume on 22 June 2019 at 02:00 UTC. Top left panel: 7.3 $\mu$m $SO_2$ total column retrievals (white-purple-green-red color scale) with 11 $\mu$m brightness temperatures plotted underneath (red triangle indicates location of Raikoke). Black line indicates CALIOP track and green highlight indicates full latitude/longitude range displayed on the bottom panel. Top right panel: 532 nm total attenuated backscatter profile averaged over the latitude/longitude range highlighted on the bottom panel. Bottom panel: 532 nm total attenuated backscatter curtain (black line indicates tropopause). (b) Same (a), but for 25 June 2019 at 14:00 UTC.

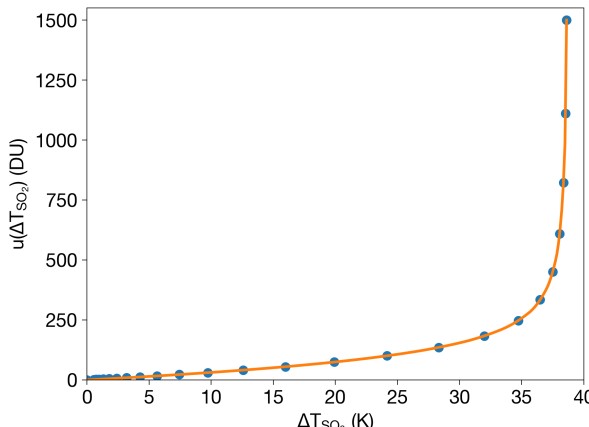

**Figure B4.** MODTRAN-6.0 simulation results used to derive $SO_2$ total column density as a function of brightness temperature difference (see Eq. B1) for a uniform distribution with cloud-top height of 13.5 km and thickness of 2.5 km.

**Table 1.** Summary of model setup for the test suite validation cases shown in this paper.

| Parameter | Puyehue-2011 (A) | Puyehue-2011 (B) | Raikoke-2019 (A) | Raikoke-2019 (B) | Etna-2013 | Chernobyl-1986 |
|---|---|---|---|---|---|---|
| Start date | 2011-06-04 | 2011-06-05 | 2019-06-21 | 2019-06-22 | 2013-02-23 | 1986-04-25 |
| Start time | 21:00 UTC | 15:00 UTC | 18:00 UTC | 18:00 UTC | 18:00 UTC | 00:00 UTC |
| Run period | 99 h | 81 h | 72 h | 48 h | 10 h | 384 h |
| Resolution (hor.) | 0.1° | 0.1° | 0.1° | 0.1° | 0.015° | 0.125° |
| Vertical levels | 60 | 60 | 80 | 80 | 60 | 60 |
| Species | Fine ash | Fine ash | $SO_2$ | $SO_2$ | Tephra | Radionuclides |
| Data insertion | No | Yes | No | Yes | No | No |
| Source type | HAT | HAT | SUZUKI | No source | HAT | HYBRID |
| Initial col. height | 11.2 km | 13 km | 15.5 km (max)[1] | 13.5 km | 8.7 km | 3.3. km |
| Initial col. thickness | 2 km | 2 km | - | 2.5 km | 3.5 km | - |
| Meteo. driver | ERA5 | ERA5 | GFS | GFS | WRF-ARW | ERA5 |
| | | | Validation strategy | | | |
| Validation data | SEVIRI (Meteosat-9) collocated with CALIPSO | | AHI (Himawari-8) collocated with CALIPSO | | 10 ground points | 56 ground points (REM database) |
| Validation metrics | SAL | | SAL | | point-to-point error | point-to-point error |

[1] Variable column height between 3.5 and 15.5 km

**Table 2.** Description of the files in the **Puyehue-1986** test suite folder (see Sec. 3.1).

| Sub-folder | File | Description |
|---|---|---|
| *InputFiles* | *puyehue-1986.inp* | *FALL3D-8.0* input file |
| | *Puyehue-2011.ash-retrievals.nc* | netCDF file with SEVIRI ash retrievals. Used for model data insertion and validation |
| *Meteo* | *download_era5_sfc_puyehue2011.py* | python script to download ERA5 surface variables in netCDF format [1] |
| | *download_era5_ml_puyehue2011.py* | python script to download ERA5 model level variables in netCDF format [1] |
| | *merge_ml_sfc_puyehue2011.sh* | shell script to merge the files above into a single netCDF file [2] |
| *Validation* | *validate_puyehue.py* | python script to validate model results. Writes SAL metrics on *validation_metrics_puyehue.txt* |
| | *vmetrics.py* | python library needed (imported) by *validate_puyehue.py* [3] |
| | *validation_metrics_puyehue_expected.txt* | expected results file. Should coincide with *validation_metrics_puyehue.txt* |

[1] Makes use of CDS API of the Copernicus Climate Data Store

[2] Makes use of Climate Data Operators (CDO)

[3] Makes use of *netCDF4*, *numpy*, *datetime*, *pandas* and *scipy* python libraries

**Table 3.** Description of the files in the **Etna-2013** test suite folder (see Sec. 3.3).

| Sub-folder | File | Description |
|---|---|---|
| *InputFiles* | *Etna-2013.inp* | *FALL3D-8.0* input file |
| | *Etna-2013.tgsd.tephra* | Total Grain Size Distribution (TGSD) file (not generated by the *SetTgsd* pre-process task) |
| | *Etna-2013.pts* | tracking points file (coordinates of the 10 ground observation locations) |
| *Meteo/ERA5* | *download_ERA5_sfc_etna2013.py* | python script to download ERA5 surface variables in grib2 format [1] |
| | *download_ERA5_ml_etna2013.py* | python script to download ERA5 model level variables in grib2 format [1] |
| *Meteo/WRF* | *namelist.wps* | **Etna-2013** input file for the WRF Pre-Processing System (WPS) |
| | *namelist.input* | **Etna-2013** input file for both the *real.exe* and *wrf.exe* executables |
| *Validation* | *validate_etna.py* | python script to validate model results. Writes deposit metrics on *validation_etna.csv* [2] |
| | *validation_etna_expected.csv* | expected results file. Should coincide with *validation_etna.csv* |

[1] Makes use of CDS API of the Copernicus Climate Data Store

[2] Makes use of *pandas*, *glob*, *os*, and *sys* python libraries

**Table 4.** Summary of the SAL and FMS validation scores for the **Puyehue-2011** and **Raikoke-2019** test suite cases. The 'DI' columns indicate validation scores for runs with data insertion and 'NDI' indicates scores for runs with no data insertion.

| Validation metrics | S | | A | | L | | \|SAL\| | | FMS | |
|---|---|---|---|---|---|---|---|---|---|---|
| | DI | NDI | DI | NDI | DI | NDI | DI | NDI | DI | NDI |
| **Puyehue-2011** | | | | | | | | | | |
| 0 h | 0.0 | -1.31 | 0.0 | -0.5 | 0.0 | 0.12 | 0.0 | 1.93 | 1.0 | 0.22 |
| 6 h | 0.04 | -1.48 | -0.24 | -0.68 | 0.11 | 0.08 | 0.39 | 2.24 | 0.56 | 0.26 |
| 12 h | -0.37 | -1.68 | -0.3 | -0.69 | 0.24 | 0.05 | 0.91 | 2.42 | 0.48 | 0.24 |
| 18 h | -0.17 | -1.57 | -0.33 | -0.68 | 0.36 | 0.07 | 0.86 | 2.31 | 0.43 | 0.29 |
| 24 h | -1.0 | -1.22 | -0.22 | -0.54 | 0.08 | 0.09 | 1.3 | 1.84 | 0.42 | 0.32 |
| 30 h | -0.6 | -0.85 | -0.24 | -0.52 | 0.1 | 0.12 | 0.95 | 1.49 | 0.38 | 0.3 |
| 36 h | -0.55 | -0.77 | -0.01 | -0.23 | 0.74 | 0.75 | 1.3 | 1.76 | 0.4 | 0.34 |
| 42 h | 0.46 | 0.15 | 0.11 | -0.06 | 0.63 | 0.37 | 1.19 | 0.58 | 0.3 | 0.28 |
| 48 h | -0.83 | -0.84 | 0.08 | 0.06 | 0.32 | 0.3 | 1.24 | 1.2 | 0.14 | 0.14 |
| 54 h | 0.04 | 0.03 | 0.41 | 0.43 | 0.71 | 0.67 | 1.16 | 1.13 | 0.11 | 0.1 |
| 60 h | 0.64 | 0.64 | 0.83 | 0.83 | 0.67 | 0.67 | 2.15 | 2.15 | 0.09 | 0.09 |
| 66 h | 0.41 | 0.41 | 0.91 | 0.91 | 0.45 | 0.45 | 1.76 | 1.76 | 0.09 | 0.09 |
| 72 h | 0.46 | 0.46 | 0.89 | 0.89 | 0.36 | 0.36 | 1.71 | 1.71 | 0.1 | 0.1 |
| **Raikoke-2019** | | | | | | | | | | |
| 0 h | 0.0 | -1.18 | 0.0 | 1.67 | 0.0 | 0.02 | 0.0 | 2.87 | 1.0 | 0.32 |
| 6 h | -0.09 | -0.98 | -0.13 | 1.63 | 0.02 | 0.04 | 0.24 | 2.65 | 0.52 | 0.28 |
| 12 h | -0.24 | -1.14 | -0.29 | 1.57 | 0.04 | 0.05 | 0.57 | 2.76 | 0.37 | 0.24 |
| 18 h | -0.46 | -0.89 | -0.46 | 1.51 | 0.04 | 0.04 | 0.96 | 2.44 | 0.31 | 0.22 |
| 24 h | -0.67 | -1.03 | -0.49 | 1.5 | 0.05 | 0.04 | 1.21 | 2.58 | 0.29 | 0.23 |
| 30 h | -0.62 | -0.82 | -0.59 | 1.45 | 0.05 | 0.04 | 1.27 | 2.31 | 0.24 | 0.22 |
| 36 h | -0.6 | -0.52 | -0.58 | 1.46 | 0.11 | 0.09 | 1.29 | 2.07 | 0.25 | 0.2 |
| 42 h | -0.84 | -0.52 | -0.69 | 1.4 | 0.07 | 0.06 | 1.6 | 1.97 | 0.23 | 0.21 |
| 48 h | -0.68 | -0.47 | -0.67 | 1.39 | 0.03 | 0.02 | 1.38 | 1.88 | 0.25 | 0.2 |

**Table 5.** Total radioactivity emitted in the atmosphere during the Chernobyl accident in the period 24 April - 10 May, 1986, for Caesium and Iodine isotopes, and their best fit fractions in the considered settling velocity classes

| Radionuclide | Total activity (PBq) | Vs=mm/s | Vs=3mm/s | Vs=4mm/s | Vs=6mm/s |
|---|---|---|---|---|---|
| $^{134}$Cs | 54 | 0.54 | 0.46 | | |
| $^{137}$Cs | 85 | | | 1.0 | |
| $^{131}$I | 1760 | | | | 1.0 |