# Peer review of "FALL3D-8.0: a computational model for atmospheric transport and deposition of particles, aerosols and radionuclides – Part 2: model validation"

_Geoscientific Model Development, 2020_

## Referee Comment (RC1) · Anonymous Referee #1 · 1 Jul 2020

General Comments: The paper provides some examples of FALL3D-8.0 simulations in an attempt to validate the model and demonstrate its capabilities for volcanic eruptions and radionuclide accidents. Whilst these are interesting examples, unfortunately the paper falls far short of its part 1 sister paper's (Folch 2020) claim that it will be "a detailed FALL3D-8.0 model validation for several simulations that are part of the new benchmark suite of the code" and it does not stand up as a validation piece for the FALL3D model. I don't believe it can be accepted in its current form as a Part2 paper.

Particular concerns are that: - There is minimal reference to the FALL3D model code

itself, which makes it questionable as to whether the paper is suitable for GMD. - It reads like 4 discrete pieces of work that have been thrown together without a coherent aim for what good validation looks like. - There is no description of the benchmark suite. In fact, the first mention of this is at the end of the paper. - There is no demonstration of the skill of FALL3D-8.0 compared to previous versions of the model - Whilst the title talks about applications, other papers have already been published using FALL3D for most of these applications so there is already far more detailed evidence in the literature for its use in these areas.

I believe the intentions of the authors are well-founded, but this current manuscript falls short. The examples neither prove the capability of the model to replicate ash or SO2 plumes (demonstrated by the poor agreement with time between the model and observations) nor demonstrate that v8 is an improvement over v7. It also doesn't provide any indication of how FALL3D-8.0 compares against other models for well-established datasets, which would be another route to provide validation of the model.

The authors potentially have two choices in my opinion: (1) to turn this into a paper that explores how new satellite retrieval techniques can be used to inform volcanic plume modelling for the Cordon-Caulle and Raikoke cases and submit elsewhere or (2) give this a major rewrite to focus on the role and construction of the benchmark suite, and how the model's performance has improved (assuming that this is shown in results) in v8 compared to v7 (or earlier versions).

Option 2 will require new simulations to be performed and the authors will need to put time aside to enable this.

I appreciate that lack of good data is a real challenge for these sorts of models, but a good model validation paper including a description of the approach to benchmarking would be valuable and so I encourage the authors to consider how this can be achieved.

I have provided some more lengthy feedback below.

Specific Comments:

It is unclear from the abstract what the purpose of the paper is – either a demonstration of the range of its applications, or the validation piece that accompanies the description of the new code in part 1. Based on lines 27-30 it seems to be validation, which is what I would agree this paper should be about, hence it would be better to structure the title and abstract around this. I would recommend removing the text "different application cases and" from Line 1 as a start.

The reference to the description of the eruption "in sec 4.1" on Line 55 is an indication that the structure of the paper could be improved. For the reader it would be preferable to describe this eruption (and the other case-studies) prior to this point if it is being used in section 2, as Section 4 is too late. I would suggest that the parts of Section 4 describing the four case-studies should come after the introduction to set the scene and to explain/justify why these examples have been chosen for the benchmarking and why they are the most pertinent to the aim of validating the model.

Given that the paper is submitted to GMD, it is surprising how many pages are devoted to explaining the derivation of satellite data (5 pages and 8 figures) compared to the actual analysis of model skill (5 pages). This doesn't seem like a good balance. Particularly as the satellite data is not relevant for two of the case-studies. The detailed description of the retrievals in 2.1.1 and 2.1.2 is not a good fit for this paper, particularly as new unpublished techniques are introduced. It would be preferable for these techniques to be introduced and peer-reviewed in a remote sensing paper/journal and then just referred to in this model validation paper. An alternative would be to move these sections to supplementary material to keep the focus on the model, but this risks these techniques not being properly documented. I appreciate that there is a balance to be struck between needing to explain the data used and the actual validation, but currently the emphasis of the paper is not right. There is also too much focus on validating the data insertion scheme rather than the model. If this is meant to be an example of using a benchmark to test different approaches to the source description (i.e. no-insertion,

insertion, other options), then this should be made much clearer.

In Fig 9, a description is needed of what the different colour plumes are in the right-hand side panels. Using different colour contours for the same threshold in the left and central panels is really quite confusing. I can see what the authors have tried to achieve (i.e. to make them match the colours in the right-hand panel), but using different colours is misleading with respect to the contour scale. I would suggest removing the coloured line contours, as the plume extent is clear, or putting both in grey/black.

For a model validation paper, this is lacking a significant amount of detail about how the runs have been conducted. For the Cordon-Caulle and Raikoke simulations details are only given in the text about the horizontal grid and the data insertion time. Information is missing on the vertical grid, model timesteps and other model parameters, as well as the meteorology used. More critically, information is almost completely absent about the emissions (eruption source parameters) including: the depth of the plume used in the insertion case, the source term that has been used in the no-insertion case, how the continued eruption of the volcano is represented in the insertion case for Cordon-Caulle, species properties, etc etc. This is crucial information to understand how the model has been run and needs to be included, ideally in a new methods section. Reference should also be made to Table 1 in each section.

I'm not really sure what the Cordon Caulle example is trying to demonstrate. The first outputs in Fig 9a and 10a (2011-06-05 15:00 UTC) are a good example of how insertion at one timestep can correct for poor source information, but as the authors state the impact of this has disappeared by 48 hours. After this time neither set of simulations appears to validate well with the satellite data and are actually similar, because presumably they are using the same (although as highlighted above) unspecified source. This example doesn't really prove either (i) the value of insertion for a long eruption or (ii) the ability of the model to represent the plume. I suggest that the author's need to think more carefully about the purpose of this case-study and the reason for including it.

For the Raikoke case, the text should highlight that FALL3D does not have a chemistry scheme and no conversion is occurring. Are any loss processes being accounted for in the simulations? The text in lines 348-356 focuses on poorly referenced speculation as to why the observed SO2 could be increasing, but no mention is given as to whether the change in score is due to the modelled SO2 mass decreasing too rapidly. This needs to be considered in the text, even if it is just to rule it out. The Cordon Caulle and Raikoke case-studies are focussed on the data insertion scheme, which is a minor component of the model. To prove that v8 has introduced any enhancement to the model itself, these examples need to include a comparison for the no-insertion cases with the previous version of FALL3D.

I was pleased to see much more detail about the simulation set-up provided for the Etna case in the text, but this is needed for each case not just this one. The level of detail in lines 375 to 389 highlights just how much information is missing from the Cordon-Caulle description. But, as with other comments, I'd suggest these details would be better in a methods section.

Table 1: what is the difference between "fine ash" and "tephra" in the model? As a minimum this should refer back to Folch et al (2020) Table 3, but it would be much clearer to have this defined in the text.

The Poret 2018 paper is a very detailed study of this Etna eruption using FALL3D and so I am struggling to see what its inclusion in this paper brings. It has already been demonstrated that FALL3D can be applied to this sort of study. This could have been a very good opportunity to compare v8 of the model against a previous study, but this is not done. The main difference appears to be the use of much higher resolution meteorology, which is an input to the model, not the model itself. Is this the reason why the agreement between the model and obs appears (at least at face value from the log graphs in both papers) to be better in this paper? Much more detail on the reason for the differences is needed and to justify the inclusion of this case-study in this paper.

[Figure]

For the Chernobyl case, I am concerned that the authors have tuned the model's settling velocities to unphysical values in order to create a better match between the model and observations, but the text is vague enough to not make this clear. If this is the case, then are the authors suggesting that these are the values that have been implemented as the defaults within FALL3D? This seems unscientific and is just tuning the model to this specific case-study. More detail is needed here.

In Figure 17, an explanation is needed as to what the dashed lines show. In addition, to fit with Fig 15, the dashed lines should really be changed from a factor of 10 to x3. There would then be the basis for a much more meaningful discussion as to why the model performs so differently in these two cases, given that both are for ground deposits, and what the causes of these differences are. This would be much more powerful and useful for the scientific community.

Remove figure 18. Without any reference data to compare it to, it is just a pretty picture.

It is clear from the style that the text in the Chernobyl case study has been written by a different co-author to the rest of the text and some grammar improvements would be useful. As with the previous case-studies this example doesn't provide any real validation of the model, as would be required in a genuine part 2 paper.

The mention of the FALL3D Benchmark Suite in the first line of the Conclusions is a complete surprise. Surely this suite should be the focus of the entire paper in that case? Why is it not the common thread throughout? This would be far more appropriate for a paper for GMD. For example, describing: How does the suite work? How has it been coded up? Does it have Known Good Outputs that model tests are compared against? How many tests are run and does it test the code itself (e.g. some degree of unit tests or bit-comparability for different versions) or just the outputs? Are all the tests run for every commit? What metrics are used to show when the model is falling outside of acceptable performance? etc

As it stands, it's impossible to work out from this paper whether FALL3D-v8 is actually

any good.

Technical Corrections:

Line 21: use of the word "aerosols" is incorrect here, as they all appear to be gases from the table in Folch et al (2020)

Line 24: Check/confirm the use of "chemical reactions" here, as Folch (2020) says that there is no chemistry as far as I can tell

Line 110: "new python implementation of the original FORTRAN". This text seems superfluous here, unless you are also providing access to the new code. A separate paper introducing this retrieval code would streamline this paper and keep it focused on the model.

Line 126: "ERA5 reanalysis data" - Please provide more details on what this and a reference

Line 134: You use CALIPSO here but refer to CALIOP in the figure description. The two need to be consistent, and you need to add an explanation in the text as to what CALIOP is.

Line 190: You need to explain why this is relevant, i.e. that CALIPSO detects aerosol not gas, so the layer is only a proxy for SO2

Within section 2.2.2 and elsewhere you use both Himwari-8 and AHI. This is confusing, please choose one and stick to it throughout. I would suggest that more people are familiar with Himawari than AHI.

Line 213: add "of" to "validation air"

Line 271: It would be helpful to repeat the time used for the data insertion here to explain the 1:1 agreement in Fig 9a.

Lines 304-323: This is all introductory material, so would fit better earlier on and separated from the results

Lines 333-334: It would be helpful to give the scores for the insertion case here for clarity, i.e. I assume the FMS is 1. It is a little biased to compare at the exact time of insertion, a comparison even an hour later when the model has actually had some influence on the insertion case would seem fairer.

Lines 345-5: The simulations indicate that the model is able to track the Himawari observations when initiated with a Himawari source, but other satellites, e.g. TropOMI, show a much larger SO2 plume on 23/24th, which in this case the data insertion approach does not capture as it is "tuned" to Himawari. This is an interesting question around whether the model should be validated against the same data source that is used for the insertion, it would be useful for the authors to comment on this in the paper.

Line 377: Can you provide some details as to how/if (for the horiz) and why (for vert and horiz) the resolutions used (i.e. resolution of 0.015deg and 60 vertical levels up to 11 km in) differ from the WRF input please.

Line 377: This is the first mention of "samples". What are these? These need to be explained earlier on.

Line 378: Please explain what Phi is, as this won't be known to readers of GMD.

Line 381: Explain what the two parameters at the start of this line are. Do they relate to coarse and fine?

Line 383-4: "and the resulting tephra ground load map is shown in Fig. 14." Resulting from what? The text needs to specify that this is 10hrs after the start time and corresponds to the end of the simulation. There is a lot of information missing here, which should be provided in a methods section, including: - Was the eruption source constant during this period? - What mass eruption rate was used? - Does this period correspond to when the measurements were taken? - Why is this different to the duration of the

[Figure]

Poret study simulation?

Line 393: "all points lie within a factor of 3 error band" - A factor of 2 would be the more conventional choice of statistic here. I suspect that the log scale is rather deceptive for the lack of agreement at the larger values! And it is the larger values that have the bigger impact and so are more important to get correct. It would be good to see this part of the graph on a linear scale if possible.

Line 394: with regard to the use of "10-3 km m-2" It would be helpful to understand the precision of the smallest values from both the obs and the model to understand the potential uncertainty at this end of the scale

Section 4.4: the grammar in this section needs some improvement

Line 399: "accident" should be "accidents"

Line 404: "estimations of such a source term is" – either needs to be "estimation" and "is" or "estimations" and "are"

Line 410-411 and Table 3: I am struggling to understand this. Firstly, the units need to be the same in the text and the table, either m/s or cm/s, for ease of interpretation. But secondly, the values in the table don't agree with the ranges in the text: 0.0005 to 0.005 m/s for 137Cs - but the table has 0.04 m/s; 0.001 to 0.02 m/s for 131I - but the table has 0.06 m/s; Is there a unit issue here? Or am I missing something?; Some more explanation is needed to make this clear.

Line 416: use of "best case" – what were the other cases?!

Line 454-5: This is too old a paper to make this type of claim for the current breed of models.

Line 456-460: This text is not relevant as the paper is not about model development. Further developments to the benchmark suite would be more appropriate.

Overall, I would suggest that there are too many figures.

---

## Short Comment (SC1) · 10 Jul 2020

We thank Reviewer 1 (R1) for their detailed review of our manuscript. Upon consideration of their suggestions we have decided to make some major revisions. We plan to cut back the satellite sections of the paper and move the details to an Appendix (as suggested by R1). We will also add a specific section detailing the benchmark suite.

---

## Referee Comment (RC2) · Anonymous Referee #2 · 14 Jul 2020

Review of the article "FALL3D-8.0: a computational model for atmospheric transport and deposition of particles, aerosols and radionuclides – Part 2: model applications" by Prata et al.

The manuscripts presents applications and evaluation of the Eulerian transport model FALL3D-v8, alongside a companion paper (Part 1), which presents the model physics and some limited verification. The applications are of SO2 and volcanic ash transport using a data insertion scheme, evaluated using satellite imagery; the transport and deposition of volcanic ash evaluated using tephra samples; and the transport and deposition of radionuclides evaluated using deposition measurements. This manuscript pertains to 'model evaluation', using the language of the journal. Quoting GMD's scope, they suggest that "where evaluation is very extensive, a separate paper focussed solely on this aspect may be submitted...typically, this comprises a comparison of the performance of different model configurations or parameterisations." In the case where the manuscript contains "substantial conclusions about geoscience rather than about models, and such papers are not suitable for submission to GMD." In its current form, in my opinion, the manuscript does not sufficiently evaluate the model to fit the scope of publication in GMD. Very little of the manuscript focuses on model evaluation – i.e. the model's ability to reproduce real world physics – with too much focus on the data used to evaluate. The paper has, however, passed the access review stage, suggesting that the topical editor has deemed the manuscript acceptable for GMD's scope. Therefore below I provide my suggestions for revisions to improve the manuscript, followed by technical comments.

1) The manuscript does not sufficiently evaluate the model physics. There are many ways that this could be done in harmony to the companion paper. For example, one evaluation could be through simulation of an ash cloud using the emissions terms (1)-(4) in Sect. 3.2.3 detailed in the companion paper. An additional important evaluation case is the effect of including the fourth-order Runge-Kutta scheme in the solving scheme. The superiority of the new aggregation scheme in v8 over that used in v7 should also be demonstrated.

2) The manuscript should include an example using emissions term (5) from Sect. 3.2.3 in the companion paper (i.e. resuspension). Desert dust would be a sensible choice if the authors wish to move the model away from being purely volcanological. This will better demonstrate dispersion from within the boundary layer.

3) A huge section of the paper is taken up by description of the satellite detection algorithms. This level of detail should not appear in the main text, which should focus on the model. No reasoning is given for using a bespoke satellite detection and retrieval algorithm here. A previously published algorithm should be used using an available data source to improve transparency. For example from SACS (https://sacs.aeronomie.be/) or some similar openly available source. This point is emphasised by the manuscript stating that (Line 321) 'it should be noted that the retrievals presented here are preliminary and require further cross-validation with other satellite retrievals'.

4) The choice of the 1986 Chernobyl accident seems an odd choice given the relative improvement in measurements during the Fukushima-Daiichi accident. This would also allow the authors to demonstrate the decay scheme for Strontium-90.

5) It is unclear why there is so much emphasis on data insertion. The paper generally reads as justification for using data insertion, which has already been shown in Wilkins et al. (2016). Either the volcanic ash or SO2 example should be dropped as a single example shows that the model is capable of data insertion.

Technical comments:

Abstract: Acronyms (i.e. SAL/FMS) should not be defined in the abstract.

Line 16: Change to '15+ year track record'

Section 2.1.: If keeping, a brief description of SEVIRI is needed.

Section 2.1.1.: I would strongly urge the authors to use an 'off-the-shelf' product, but if keeping then it needs to be made clear that this is a bespoke algorithm relevant to this test case.

Eq 1: The subscript 'ash' should not be italic and it would be clearer is Twc was replaced with -0.5K in the equation.

Eq 3: Place -2K in the equation

Section 2.1.2, Eq 5: What geometric thickness of the cloud is assumed here? Is this the same thickness that is used in the insertion scheme?

Eq 8: Put -2.5 K in equation

Line 172: Specify what you mean by 'meteorological clouds', i.e. water and ice clouds.

Line 178: "As mentioned above", specify the section.

Line 186: Which gases?

Line 187: What 'amounts' are you referring to?

Line 203: Is this vertical distribution also the same slab as used in the satellite retrieval? The Puyehue-Cordón Caulle eruption was known to have complex multi-layered cloud structures. How has this been dealt with in the satellite retrieval and insertion?

Lines 212 and 216: FMS and SAL need defining in their first appearance in the main body of text.

Section 3: These validation metrics are only valid for the data insertion scheme. Please provide metrics for the other test cases. How S, A and L are combined into a single metric needs to be detailed.

Line 255: Are the 'ash mass loading areas' the areas of the satellite pixels or the meteorological/output resolution of the model? How are the alternate resolutions compared?

Sections 4.1 and 4.2: These section are evaluating the data insertion method, which has been evaluated in previous work, rather than the model itself.

Section 4.1.: What is the grain size distribution used in this case? Assuming it is the retrieved effective radius, how many bins are used etc?

Lines 259-263 and 264-269 can be cut. This information is superfluous to the model.

Line 299: Change 'meteorological clouds' to relevant cloud type.

Lines 206-310 can be cut.

Line 370: It is not clear to me why ARW was run first. Why was this initial step needed?

Line 393: 'factor 3 error band' should just be 'a factor of 3'.

Line 399: 'nuclear accidents'

Line 4.4: In contract to the lengthy explanation of the observations in the other test cases, nowhere in this section does it specify what was actually measured. This section also needs discussion on how general this set up is. For example, the FALL3Dv8 would be unable to be used to model the recent 2017 release of ruthenium-106 in Europe nor iodine-135/xenon-135 during Fukushima.

Line 416: The supplementary material contains nothing on how it accounts for diffusion, deposition nor decay.

Section 5: Many of the conclusions are about the satellite detection scheme and applications of the model, rather than evaluation of the model itself.

Line 456-459: Model performance has not been discussed anywhere else in the manuscript and therefore does not serve as a conclusion/future work. This should be removed unless performance is explicitly detailed elsewhere in the manuscript.

–––––––––––––––––––––––––––

---

## Author Comment (AC1) · 10 Sep 2020

**Response to R1**

*R1 general comments*

1. The paper provides some examples of FALL3D-8.0 simulations in an attempt to validate the model and demonstrate its capabilities for volcanic eruptions and radionuclide accidents. Whilst these are interesting examples, unfortunately the paper falls far short of its part 1 sister paper's (Folch 2020) claim that it will be "a detailed FALL3D-8.0 model validation for several simulations that are part of the new benchmark suite of the code" and it does not stand up as a validation piece for the FALL3D model. I don't believe it can be accepted in its current form as a Part2 paper. Particular concerns are that:
   - There is minimal reference to the FALL3D model code itself, which makes it questionable as to whether the paper is suitable for GMD.
   - It reads like 4 discrete pieces of work that have been thrown together without a coherent aim for what good validation looks like.
   - There is no description of the benchmark suite. In fact, the first mention of this is at the end of the paper.
   - There is no demonstration of the skill of FALL3D-8.0 compared to previous versions of the model.
   - Whilst the title talks about applications, other papers have already been published using FALL3D for most of these applications so there is already far more detailed evidence in the literature for its use in these areas.

We thank the reviewer for these points. We realised that while it is true that, although all the information is already in the manuscript, the test suite of the code and the novelties introduced in the code with respect to the previous versions were not properly described. In the substantially revised manuscript we now:
   - Describe in detail the test suite (a completely new section has been added for this purpose)
   - The testsuite is available on a GitLab repository, in which all files to run and validate the model are provided. This will allow any user/reader to exactly reproduce all the simulations/results in the paper.
   - The former Section 2 (satellite retrievals) has been radically cut down in the manuscript and moved to 2 new Appendices. In this way, the manuscript now is articulated only around the test suite, which is actually the purpose of the paper.
   - We have made it more clear that the model has never been used for data insertion, $SO_2$ dispersion and radionuclide modelling.
   - The performance of FALL3D-8.0 model for the 2013 Etna deposition case is now properly compared with the results obtained with the previous code version and the improvements highlighted.
   - We have also made great efforts to homogenise the manuscript in terms of writing style as well as structure and layout of the various sections.

2. I believe the intentions of the authors are well-founded, but this current manuscript falls short. The examples neither prove the capability of the model to replicate ash or SO2 plumes (demonstrated by the poor agreement with time between the model and observations) nor demonstrate that v8 is an improvement over v7. It also doesn't provide any indication of how FALL3D-8.0 compares against other models for well established datasets, which would be another route to provide validation of the model.

We think that these issues are due to the poor organisation of the original manuscript. Probably it hasn't been adequately highlighted that we use validation metrics to objectively show how the model is able to reproduce $SO_2$ and ash plumes (e.g. SAL scores of less than 2 out to 48 h). We have emphasised this by restructuring the manuscript with specific sections on the validation datasets and model setups for each case study. We have also added dedicated sections on the validation metrics and validation results. The route we have chosen for validation is comparison with satellite retrievals. This work (and associated validation datasets) now form the basis for future model inter-comparison studies. We have added this comment to the conclusions section.

3. The authors potentially have two choices in my opinion: (1) to turn this into a paper that explores how new satellite retrieval techniques can be used to inform volcanic plume modelling for the Cordon-Caulle and Raikoke cases and submit elsewhere or (2) give this a major rewrite to focus on the role and construction of the benchmark suite, and how the model's performance has improved (assuming that this is shown in results) in v8 compared to v7 (or earlier versions). Option 2 will require new simulations to be performed and the authors will need to put time aside to enable this. I appreciate that lack of good data is a real challenge for these sorts of models, but a good model validation paper including a description of the approach to benchmarking would be valuable and so I encourage the authors to consider how this can be achieved. I have provided some more lengthy feedback below.

We thank the reviewer for their suggestions. Upon consideration of these suggestions we decided to give the paper a major rewrite (i.e. option (2) of the reviewers suggestions). The focus of the paper is now on the test suite of the code. We also discuss how v8 has improved since v7 for the Mt Etna 2013 case study as we are able to make comparisons for this. Reasons for not comparing v7 and v8 for the other case studies are detailed in our response below.

**R1 specific comments**

1.  It is unclear from the abstract what the purpose of the paper is – either a demonstration of the range of its applications, or the validation piece that accompanies the description of the new code in part 1. Based on lines 27-30 it seems to be validation, which is what I would agree this paper should be about, hence it would be better to structure the title and abstract around this. I would recommend removing the text "different application cases and" from Line 1 as a start.

    Thank you for raising these points. The paper is about the test suite and validation of the FALL3D model. As suggested we have re-written the Abstract to reflect this. We have also revised the title to: "FALL3D-8.0: a computational model for atmospheric transport and deposition of particles, aerosols and radionuclides – Part 2: model validation".

2.  The reference to the description of the eruption "in sec 4.1" on Line 55 is an indication that the structure of the paper could be improved. For the reader it would be preferable to describe this eruption (and the other case-studies) prior to this point if it is being used in section 2, as Section 4 is too late. I would suggest that the parts of Section 4 describing the four case-studies should come after the introduction to set the scene and to explain/justify why these examples have been chosen for the benchmarking and why they are the most pertinent to the aim of validating the model.

    We agree. We have completely restructured the manuscript so that the descriptions of the case studies come before the descriptions of the validation datasets and model setups for each case. In the descriptions of the case studies we have provided motivation/reasons for including the case studies in the FALL3D-8.0 test suite.

3.  Given that the paper is submitted to GMD, it is surprising how many pages are devoted to explaining the derivation of satellite data (5 pages and 8 figures) compared to the actual analysis of model skill (5 pages). This doesn't seem like a good balance. Particularly as the satellite data is not relevant for two of the case-studies. The detailed description of the retrievals in 2.1.1 and 2.1.2 is not a good fit for this paper, particularly as new unpublished techniques are introduced. It would be preferable for these techniques to be introduced and peer-reviewed in a remote sensing paper/journal and then just referred to in this model validation paper. An alternative would be to move these sections to supplementary material to keep the focus on the model, but this risks these techniques not being properly documented. I appreciate that there is a balance to be struck between needing to explain the data used and the actual validation, but currently the emphasis of the paper is not right. There is also too much focus on validating the data insertion scheme rather than the model. If this is meant to be an example of using a benchmark to test different approaches to the source description (i.e. no-insertion, insertion, other options), then this should be made much clearer.

    We agree with the reviewer that the originally submitted manuscript was too heavy on the side satellite observations. However, we note that the retrieval methods used are not new and only differ in the detail of the implementation.

In response to this comment, we have moved the bulk of the satellite retrieval descriptions to Appendices along with the relevant figures as we believe it is still important that the modelling community of GMD should be aware of what uncertainties and assumptions are involved when carrying out satellite retrievals. We also provide the relevant citations for the satellite retrievals adopted in the present study. We have also restructured the manuscript so that it is easier for the reader to understand that we are validating both the model's data insertion scheme as well as simulations without data insertion. This should be made clear in the new 'model setup' sections of the revised manuscript where we specify the two different model configurations we are testing (see, for example, Sects. 3.1.2 and 3.2.2 of the revised manuscript).

4.  In Fig 9, a description is needed of what the different colour plumes are in the right hand side panels. Using different colour contours for the same threshold in the left and central panels is really quite confusing. I can see what the authors have tried to achieve (i.e. to make them match the colours in the right-hand panel), but using different colours is misleading with respect to the contour scale. I would suggest removing the coloured line contours, as the plume extent is clear, or putting both in grey/black.

Thank you for pointing this out. We have described the different colour plumes on the right-hand side panels and have removed the coloured contours as suggested (see Figs. 1, 2, 4 and 5 of the revised manuscript).

5.  For a model validation paper, this is lacking a significant amount of detail about how the runs have been conducted. For the Cordon-Caulle and Raikoke simulations details are only given in the text about the horizontal grid and the data insertion time. Information is missing on the vertical grid, model timesteps and other model parameters, as well as the meteorology used. More critically, information is almost completely absent about the emissions (eruption source parameters) including: the depth of the plume used in the insertion case, the source term that has been used in the no-insertion case, how the continued eruption of the volcano is represented in the insertion case for CordonCaulle, species properties, etc etc. This is crucial information to understand how the model has been run and needs to be included, ideally in a new methods section. Reference should also be made to Table 1 in each section.

Thank you for this comment. To address the reviewer's concerns, we now provide detailed descriptions of the model setup (in the body of the manuscript) for each case study (see Sects. 3.1.2, 3.2.2, 3.3.2 and 3.4.2 of the revised manuscript). We also refer to Table 1 in each of these subsections. These new descriptions should help to clarify the points raised in this reviewer comment. In addition, ALL files are now available in the test suite GitLab repository.

6. I'm not really sure what the Cordon Caulle example is trying to demonstrate. The first outputs in Fig 9a and 10a (2011-06-05 15:00 UTC) are a good example of how insertion at one timestep can correct for poor source information, but as the authors state the impact of this has disappeared by 48 hours. After this time neither set of simulations appears to validate well with the satellite data and are actually similar, because presumably they are using the same (although as highlighted above) unspecified source. This example doesn't really prove either (i) the value of insertion for a long eruption or (ii) the ability of the model to represent the plume. I suggest that the author's need to think more carefully about the purpose of this case-study and the reason for including.

Thank you for raising these points. First we would like to clear up the reviewer's confusion about the difference between the simulations with and without data insertion. After 48 hours the simulations are expected to be almost the same as almost all of the data inserted ash has exited the domain (this was stated on lines 289-290 of the original manuscript). The reason why this is expected is that after the data insertion time the sources in the data insertion case and the no data insertion case are the same. We apologise if this was not clear in the original manuscript. To address this we have added a more detailed description of the model setups in the revised manuscript (see Sect. 3.1.2 of the revised manuscript). We thought very carefully about selecting the Puyehue-Cordon Caulle case before deciding to use it as a validation case study. We provide our rebuttals to this reviewer comment below:

1) The Puyehue-Cordon Caulle eruption is an exceptional example of long-range fine ash transport. It has been widely studied and observed by numerous satellite instruments and provides an extremely valuable case study for understanding long-range volcanic ash transport. No other eruption like this one has occurred in the past 10 years.

2) We have demonstrated how the model performs for very long volcanic ash-rich eruptions (72 hours or more). To show this we have provided quantitative, objective validation metrics (see Fig. 11 of the original manuscript/Fig. 3 of the revised manuscript) and have found that the SAL metric remains below 2 (out of a possible worst score of 6) for almost the entire duration of the simulation.

3) We have demonstrated how data insertion corrects for poor source information. The data insertion step is done 18 hours after the simulation without data insertion. We consider 18 hours to be a long time after the eruption began. Indeed, the reviewer has stated that Fig 9a and 10a provide a clear demonstration of why data insertion is valuable.

7. For the Raikoke case, the text should highlight that FALL3D does not have a chemistry scheme and no conversion is occurring. Are any loss processes being accounted for in the simulations? The text in lines 348-356 focuses on poorly referenced speculation as to why the observed SO2 could be increasing, but no mention is given as to whether the change in score is due to the modelled SO2 mass decreasing too rapidly. This needs to be considered in the text, even if it is just to rule it out. The Cordon Caulle and Raikoke case-studies are focussed on the data insertion scheme, which is a minor component of the model. To prove that v8 has introduced any enhancement to the model itself, these examples need to include a comparison for the no-insertion cases with the previous version of FALL3D.

We thank the reviewer for raising these points - please see our responses below.

1) $SO_2$ loss processes. As suggested, we have now explicitly stated that there is no $SO_2$ chemistry in FALL3D-8.0. The only $SO_2$ loss processes considered in FALL3D-8.0 are the deposition mechanisms meaning that it is unlikely that the model is losing mass too rapidly (see Section 5 of the revised manuscript).

2) $SO_2$ mass increasing in the satellite retrievals. This is a fact not a possibility. We do observe an increase in mass in the satellite retrievals. There are several reasons related to the detection sensitivity of the retrieval (in addition to ice-$SO_2$ sequestration) as to why this can happen. We already stated these reasons in the original manuscript. In the revised manuscript we have provided more references to support these claims. See Section 5 of the revised manuscript.

3) Comparisons with the previous version of FALL3D. First the reviewer must understand that the data insertion scheme is only possible in FALL3D-8.0 and so it is impossible to compare data insertion simulations for the current and previous model versions for ash or $SO_2$. Second, the previous version of FALL3D cannot do $SO_2$/radionuclides simulations and so 'no data insertion' comparisons with the previous versions of FALL3D for Raikoke and Chernobyl test cases are also impossible. In addition, ERA5 and the current GFS dataset cannot be read by previous versions of FALL3D. Finally, this only leaves the option of comparing current and previous versions of FALL3D for 'no data insertion' simulations of volcanic ash using the Etna test case, which uses WRF-ARW as meteorological input data. FALL3D-8.0 uses a high-resolution Kurganov–Tadmor (KT) scheme combined with a fourth-order explicit Runge–Kutta which leads to a better accuracy than the solver used in previous versions of FALL3D, based on the Lax–Wendroff (LW) central-difference scheme and a first-order Euler time-marching method. A comparative analysis between both solvers can be found in the companion paper (Part 1) for ideal cases and we do not consider it necessary to include a new comparative analysis here. Major improvements of version 8.0 are performance and scalability of the code and memory management, as discussed in Part 1.

8. I was pleased to see much more detail about the simulation set-up provided for the Etna case in the text, but this is needed for each case not just this one. The level of detail in lines 375 to 389 highlights just how much information is missing from the CordonCaulle description. But, as with other comments, I'd suggest these details would be better in a methods section.

We agree and have added all the model input details in their own subsections attached to each case study in the revised manuscript.

9. Table 1: what is the difference between "fine ash" and "tephra" in the model? As a minimum this should refer back to Folch et al (2020) Table 3, but it would be much clearer to have this defined in the text.

In FALL3D, the "TEPHRA" class is subdivided into 4 subcategories, depending on particle size:
- Fine ash: $d \leq 64\ \mu m$
- Coarse ash: $64\ \mu m < d \leq 2\ mm$
- Lapilli: $2\ mm < d \leq 64\ mm$
- Bomb: $64\ mm < d$

This has now been clarified in the Introduction section of the revised manuscript as follows:

"In FALL3D, "tephra" species are subdivided into four subcategories, depending on particle diameter, $d$ (Folch et al., 2020): (i) fine ash ($d \leq 64\ \mu m$), (ii) coarse ash ($64\ \mu m < d \leq 2\ mm$), (iii) lapilli ($2\ mm < d \leq 64\ mm$), and (iv) bomb ($d > 64\ mm$)."

10. The Poret 2018 paper is a very detailed study of this Etna eruption using FALL3D and so I am struggling to see what its inclusion in this paper brings. It has already been demonstrated that FALL3D can be applied to this sort of study. This could have been a very good opportunity to compare v8 of the model against a previous study, but this is not done. The main difference appears to be the use of much higher resolution meteorology, which is an input to the model, not the model itself. Is this the reason why the agreement between the model and obs appears (at least at face value from the log graphs in both papers) to be better in this paper? Much more detail on the reason for the differences is needed and to justify the inclusion of this case-study in this paper.

The main differences between our study and Poret et al. (2018) are: (i) the horizontal resolution of both met data and dispersal model, (ii) the vertical coordinate system, (iii) the Total Grain Size Distribution (TGSD) used to define the source term, and (iv) the dry deposition parameterization. Numerical diffusion errors are reduced here because we performed higher resolution than Poret et al. (2018), which is manifested in an ash loading field extended over a larger area in Poret (2018) (compare Fig. 14 in our work to Fig. 6 in Poret et al., 2018). To show this we performed a new simulation of the Etna eruption using a previous version of FALL3D (version v7.3). To make fair comparisons, the horizontal resolution (0.015º), the number of vertical levels (60), and the TGSD were set to be consistent with the simulations performed with FALL3D v8.0. Results are shown in Fig. 1 and Fig. 2 below. Figure 1 compares the 10E-4 kg/m2 contours of tephra loading according to version v7.3 (solid red line) and version v8.0 (solid black line). The new version v8.0 tends to have less numerical diffusion.

[Figure]

**Figure 1:** Tephra loading contours according to FALL3D version v7.3 (solid red line) and version v8.0 (solid black line). The new version v8.0 shows a less diffusive distribution.

A comparison between modelled and measured deposit mass loading is shown in Fig. 2 using data from 10 sampling sites. The effect of numerical diffusion is more important for fine ash, and the difference between version v7.3 and v8.0 are more evident for the distal sampling site (the lowest measured deposit load).

[Figure]

**Figure 2:** Comparison between modelled and observed deposit mass loading from 10 sampling sites. Results according to FALL3D version v8.0 (red circles) and version v7.3 (blue triangles) are shown.

Finally, note that comparisons with previous versions of FALL3D only can be performed using the Etna case. New meteorological data used in Cordon Caulle and Raikoke test cases cannot be ingested in FALL3D v7.3. Similarly, radionuclide modelling (required for the Chernobyl case) is a new feature of the model. We have added more detail and discussion on the differences between our study and the Poret et al. study in the revised manuscript also comparing the RMSE values. Please see Section 5.3 of the revised manuscript.

11. For the Chernobyl case, I am concerned that the authors have tuned the model's settling velocities to unphysical values in order to create a better match between the model and observations, but the text is vague enough to not make this clear. If this is the case, then are the authors suggesting that these are the values that have been implemented as the defaults within FALL3D? This seems unscientific and is just tuning the model to this specific case-study. More detail is needed here.

We thank the reviewer for highlighting this point because the way in which settling velocities were calibrated was not clearly described in the originally submitted manuscript. Thanks to this comment we found a typo in Table 3 of the original manuscript, which had incorrectly reported the setting velocities in cm/s instead of mm/s. This was the source of confusion leading to the reviewer's assertion that settling velocities were "unphysical". This typo has been corrected in our revision (see Table 5 of the revised manuscript).  After this correction it is easy to recognise that our best-fitted values of the isotopes' settling velocities lie within the empirical ranges that characterise the radionuclide species we studied. Moreover, for an easier comparison, in the text, we have also converted the settling velocities reported by Brandt et al. (2002) to mm/s. A brief description of the best-fit procedure (based on grid-search method) is reported in the revised manuscript (see Sect. 3.4.2).

12. In Figure 17, an explanation is needed as to what the dashed lines show. In addition, to fit with Fig 15, the dashed lines should really be changed from a factor of 10 to x3. There would then be the basis for a much more meaningful discussion as to why the model performs so differently in these two cases, given that both are for ground deposits, and what the causes of these differences are. This would be much more powerful and useful for the scientific community.

A description of the meaning of the solid and dashed lines has been added in Fig. 9 of the revised manuscript (Fig. 17 of the original submission). Concerning the comparison of the performances between Etna and Chernobyl simulations, we think that the comparison of the Etna case (tephra fallout at short distances) with the Chernobyl case (decaying radionuclides at continental distances) could be not completely appropriate.

13. Remove figure 18. Without any reference data to compare it to, it is just a pretty picture.
We disagree with the reviewer. Fig. 18 (now Fig. 10 of the revised manuscript) shows that the 1986 Chernobyl simulations correctly reproduce the patterns described by Brandt et al. (2002). The Brandt et al. study provides a good reference for comparison. Also we have provided the REM data in the FALL3D-8.0 test suite GitLab repository. We have therefore chosen to retain the figure.

14. It is clear from the style that the text in the Chernobyl case study has been written by a different co-author to the rest of the text and some grammar improvements would be useful. As with the previous case-studies this example doesn't provide any real validation of the model, as would be required in a genuine part 2 paper.

In our revised submission we have made significant efforts to homogenous the writing style throughout the manuscript. In response to the reviewer's second point "As with the previous case-studies this example doesn't provide any real validation of the model", we now provide a better description of the best fitting method and values of statistical indicators such as the RMSE (see Sect. 4.3 of the revised manuscript for example). However, we would like to add that it is patently untrue that we do not provide "any real validation of the model" for the ash and $SO_2$ simulations (i.e. we use the FMS and SAL scores for these cases). We leave it to the Editor to decide whether or not this reviewer comment is reasonable given the evidence.

15. The mention of the FALL3D Benchmark Suite in the first line of the Conclusions is a complete surprise. Surely this suite should be the focus of the entire paper in that case? Why is it not the common thread throughout? This would be far more appropriate for a paper for GMD. For example, describing: How does the suite work? How has it been coded up? Does it have Known Good Outputs that model tests are compared against? How many tests are run and does it test the code itself (e.g. some degree of unit tests or bit-comparability for different versions) or just the outputs? Are all the tests run for every commit? What metrics are used to show when the model is falling outside of acceptable performance?

This has been addressed by the new FALL3D-8.0 test suite section (see Section 2 of the revised manuscript) and the test suite repository on GitLab (see reply to General comments 1).

16. As it stands, it's impossible to work out from this paper whether FALL3D-v8 is actually any good.

We believe we have addressed this in our responses to the reviewer's above comments.

**R1 technical corrections**

Line 21: use of the word "aerosols" is incorrect here, as they all appear to be gases from the table in Folch et al (2020)
Accepted. We have corrected "aerosols" to "gases".

Line 24: Check/confirm the use of "chemical reactions" here, as Folch (2020) says that there is no chemistry as far as I can tell
In future versions of FALL3D we plan to add chemical reactions. Having independent sets of bins allows for this implementation along with other possibilities (like new physics parameterisations *etc*). Accordingly, we have reworded the sentence to: "These different categories and sub-categories of species can be simulated using independent sets of bins *that allow for* dedicated parameterisations for physics, emissions (source terms) and interactions among bins (*e.g.* aggregation, chemical reactions, radioactive decay *etc*)". We have also made it clear in our revisions that $SO_2$ chemistry has not been included yet (see Section 5.2 of the revised manuscript for example).

Line 110: "new python implementation of the original FORTRAN". This text seems superfluous here, unless you are also providing access to the new code. A separate paper introducing this retrieval code would streamline this paper and keep it focused on the model.
We agree and have removed this sentence as the differences in the results between the python and FORTRAN implementations are very small (for example, a one-to-one comparison of the two codes returned an $R^2$ correlation of 0.9993).

Line 126: "ERA5 reanalysis data" - Please provide more details on what this and a reference
We have provided references to the ERA5 reanalysis dataset (which is a very well-known, standard meteorological reanalysis dataset in atmospheric science) throughout the revised manuscript, along with its resolution and the number of model levels used (see Section 3.1.2 of the revised manuscript).

Line 134: You use CALIPSO here but refer to CALIOP in the figure description. The two need to be consistent, and you need to add an explanation in the text as to what CALIOP is.
CALIPSO and CALIOP are two different things. CALIPSO refers to the satellite, CALIOP refers to the lidar that is onboard CALIPSO. In fact, CALIPSO actually carries two more instruments: the IIR and the WFC. So it does make sense to refer to "CALIPSO overpasses". In the figure captions we're talking about the actual lidar measurements, so we refer to "CALIOP" there. To clarify this, we have defined the acronym "CALIOP" at its first appearance in the revised manuscript. We have also gone through the manuscript carefully to make sure that each use of these terms make sense according to the above description.

Line 190: You need to explain why this is relevant, i.e. that CALIPSO detects aerosol not gas, so the layer is only a proxy for SO2 Within section 2.2.2 and elsewhere you use both Himwari-8 and AHI. This is confusing, please choose one and stick to it throughout. I would suggest that more people are familiar with Himawari than AHI.

We have now explicitly stated that **CALIOP** does not detect gas (see Section 3.2.2 of the revised manuscript):

"We note that CALIOP total attenuated backscatter measurements are not sensitive to $SO_2$ gas and so we make the assumption that $SO_4^{2-}$ aerosols were collocated with $SO_2$ in order to assess the likely heights and thicknesses of the Raikoke $SO_2$ clouds (Carboni et al., 2016; Prata et al., 2017)."

With regard to the AHI/Himawari-8 question. With respect, this is not a matter of choice/preference. The two words have different meanings just like in the CALIOP/CALIPSO case. AHI is the instrument and Himawari-8 is the satellite that carries AHI onboard. To clear up any confusion, we have added more details about the AHI instrument and have been explicit that it is the instrument aboard the Himawari-8 satellite. Please see Section 3.2.1 of the revised manuscript.

Line 213: add "of" to "validation air"
Done. Thank you.

Line 271: It would be helpful to repeat the time used for the data insertion here to explain the 1:1 agreement in Fig 9a.
Done. The revised sentence is (see Section 5.1 of the revised manuscript):
"Figures 1 and 2 compare satellite retrievals and model simulations for the **Puyehue-2011** case at the data insertion time (15:00 UTC on 5 June 2011) as well as 24, 48 and 72 h after the insertion time for runs with and without data insertion, respectively."

Lines 304-323: This is all introductory material, so would fit better earlier on and separated from the results.
Agreed. In our revision, we have moved all case study descriptions to separate subsections that now come before the descriptions of the satellite datasets and model setups in each case (see also response to General Comment 2).

Lines 333-334: It would be helpful to give the scores for the insertion case here for clarity, i.e. I assume the FMS is 1. It is a little biased to compare at the exact time of insertion, a comparison even an hour later when the model has actually had some influence on the insertion case would seem fairer.

We have added the following sentence to the revised manuscript (Section 5.1) to avoid any confusion: "Note that for simulations with data insertion at the data insertion time the validation metrics reflect perfect scores (*i.e.* SAL = 0 and FMS = 1)". The original discussion was to emphasise how much the data insertion has an effect (in terms of quantitative validation metrics) on the simulations without data insertion. In our original submission, we were very transparent about how the model deviates from the observations after the data insertion time. In fact, we provided a table (see Table 2 of the original manuscript) showing the scores every 6 hours and also a time-series figure (Figure 11 of the original manuscript) showing the scores every hour after the insertion times. In addition to the figures that show spatially how the model deviates from the observations with time in 24 h steps (Figs. 9, 10, 12 and 13 of the original manuscript), we also provided full animations (at 1 h resolution) as Supplementary Material. We hope these points highlight the fact that our analysis is not "biased" as the reviewer seems to be claiming.

Lines 345-5: The simulations indicate that the model is able to track the Himawari observations when initiated with a Himawari source, but other satellites, e.g. TropOMI, show a much larger SO2 plume on 23/24th, which in this case the data insertion approach does not capture as it is "tuned" to Himawari. This is an interesting question around whether the model should be validated against the same data source that is used for the insertion, it would be useful for the authors to comment on this in the paper.

Thank you for raising this point. It is something that we also noticed and discussed. As suggested, we have revised the relevant discussion to:

"TROPOMI observations of the $SO_2$ cloud confirm this spatial structure (see Fig. 13 of Global Volcanism Program, 2019) and highlight the importance of understanding the limitations of the AHI retrievals used for data insertion in the present study. The reason for the lack of detection of $SO_2$ in this region in the AHI retrievals is probably due to water vapour interference, implying that this part of the plume was at lower altitudes than the main $SO_2$ cloud. Indeed, $SO_2$ height retrievals from CrIS data show that plume heights varied from ~3-7 km a.s.l. in this region (see Fig. 5 of Hyman and Pavolonis 2020)."

We have also added the following statement to the Conclusions section of the revised manuscript: "Limitations of the observations should also be taken into account when initialising simulations with data insertion."

Line 377: Can you provide some details as to how/if (for the horiz) and why (for vert and horiz) the resolutions used (i.e. resolution of 0.015deg and 60 vertical levels up to 11 km in) differ from the WRF input please.

WRF-ARW was configured with a horizontal resolution of 4 km and 100 vertical levels up to 50 hPa (i.e., ~20 km above sea level). We used hourly ERA5 reanalysis data with spatial resolution of approximately 0.25º×0.25º and 137 vertical levels from the surface up to a height of 80km, obtained from the ECMWF (European Center for Medium-range Weather Forecasts) (Dee et al., 2011) as initial meteorological field and boundary condition for the WRF domain.

As a consequence, the meteorological fields were downscaled from ~25 km (0.25º, ERA5) to 4 km (WRF-ARW), representing a 6:1 ratio, approximately. Typically, the rule is not to exceed a ratio of 7:1. So, our criterion was to increase WRF resolution relative to ERA5 resolution, but not to exceed this maximum ratio. We further increased the FALL3D resolution with respect to the WRF-ARW model by setting 0.015º (~2 km). In this case, the limitation is given by the stability condition of the dispersal model. If the cell size is too small, increasingly small times are required in order not to violate the CFL condition. Therefore, the maximum resolution of FALL3D was limited by the available computational resources. In summary, the FALL3D resolution was constrained by the WRF resolution and the available computational resources. In turn, the WRF resolution was limited by the driver data (i.e., ERA5 resolution).

Line 377: This is the first mention of "samples". What are these? These need to be explained earlier on. Line 378: Please explain what Phi is, as this won't be known to readers of GMD.

More details on the samples and the Phi scale are provided in the new version of the manuscript. See Section 3.3.1 of the revised manuscript.

Line 381: Explain what the two parameters at the start of this line are. Do they relate to coarse and fine?

Yes, they are related to coarse and fine fractions. We assumed a bi-Gaussian TGSD with mean and standard deviation given by $\mu_c$ and $\sigma_c$, respectively, for the coarse subpopulation and $\mu_f$ and $\sigma_f$ for the fine subpopulation. This is properly clarified in the revised manuscript (Section 3.3.2).

Line 383-4: "and the resulting tephra ground load map is shown in Fig. 14." Resulting from what? The text needs to specify that this is 10hrs after the start time and corresponds to the end of the simulation. There is a lot of information missing here, which should be provided in a methods section, including:
- Was the eruption source constant during this period?
- What mass eruption rate was used?
- Does this period correspond to when the measurements were taken?
- Why is this different to the duration of the Poret study simulation?

Resulting from the entire deposit process. Tephra mass loading increases up to an asymptotic value. For example, 99.9% of the emitted mass was deposited for an elapsed time of 8.6 h after the start of the simulation.

As a result, the final tephra ground load map will be the same, regardless of simulation time period T (if T > 9 h). To avoid confusion, we replace "and the resulting tephra ground load map" with: "and the final tephra ground load map".

In addition, we have clarified that the total time of the simulation was 10 h and provide more information on the emission source term definition:

"We considered a constant mass flow rate ( $3.814 \times 10^5 \ kg/s$ ) between 18:15 and 19:18 UTC on 23 February 2013". See Section 3.3.2 of the revised manuscript.

Line 393: "all points lie within a factor of 3 error band" - A factor of 2 would be the more conventional choice of statistic here. I suspect that the log scale is rather deceptive for the lack of agreement at the larger values! And it is the larger values that have the bigger impact and so are more important to get correct. It would be good to see this part of the graph on a linear scale if possible.

All points lie within the 1:3 ratio band and the factor 3 seems to be more significant. Not all points are within the 1:2 band. So, we see no reason to change this figure. We provide below the linear scale version of this figure, but it is not included in the manuscript:

[Figure]

Note that proximal sampling sites S1-S5, associated with the highest values of deposit load, are located <10 km from the vent. For this paper, the horizontal resolution was 0.015º (~2 km). As a result, extremely high resolution should be carried out to capture variations of tephra loading measured from proximal samples and large errors are expected in this case. For this reason, we prefer to emphasize results related to distal and medial sites (S5-S10) by using a log scale.

Line 394: with regard to the use of "10-3 kg m-2" It would be helpful to understand the precision of the smallest values from both the obs and the model to understand the potential uncertainty at this end of the scale

A tephra mass loading of 1E-3 kg/m$^2$ corresponds to a deposit thickness of 1 μm (assuming a deposit bulk density of 1000 kg/m$^3$). From an observational point of view, although difficult, as order of magnitude, studies of cryptotephra commonly detect particles up to micron (e.g., Smith et al., 2020). From a computational point of view there isn't any problem to identify a 1 micron deposit. So the use of 1E-3 kg/m$^2$ as a lower limit is justified.

Reference:
Smith V.C., Costa A., Aguirre-Díaz G., Pedrazzi D., Scifo A., Plunkett G., Poret M., Tournigand P.Y., Miles D., Dee M., McConnell J.R., Sunyé-Puchol I., Dávila Harris P., Sigl. M., Pilcher J.R., Chellman N., Gutiérrez E. (2020) The magnitude and impact of the 431 CE Tierra Blanca Joven eruption of Ilopango, El Salvador, Proc. Natl. Acad. Sci. USA, accepted.

Section 4.4: the grammar in this section needs some improvement
In our revision we have gone over and corrected for grammatical errors. In the revised manuscript, the relevant Sections are 3.4 and 5.4.

Line 399: "accident" should be "accidents"
Done.

Line 404: "estimations of such a source term is" – either needs to be "estimation" and "is" or "estimations" and "are"
Done. Changed to "are".

Line 410-411 and Table 3: I am struggling to understand this. Firstly, the units need to be the same in the text and the table, either m/s or cm/s, for ease of interpretation. But secondly, the values in the table don't agree with the ranges in the text: 0.0005 to 0.005 m/s for 137Cs - but the table has 0.04 m/s; 0.001 to 0.02 m/s for 131I - but the table has 0.06 m/s; Is there a unit issue here? Or am I missing something?; Some more explanation is needed to make this clear.
We thank the reviewer for noticing these inconsistencies. The mistakes with the units are now corrected in the revised manuscript. We also found a typo in the header of Table 3: the units are mm/s instead of cm/s. For an easier comparison between the simulations and the measured values, in our revision, we have converted the values reported by Brandt et al. (2002) to mm/s.

Line 416: use of "best case" – what were the other cases?!
We mean the "best-fitted" case. We have corrected this in the revised manuscript and added more detail about the fitting procedure (see Section 3.4.2 of the revised manuscript).

Line 454-5: This is too old a paper to make this type of claim for the current breed of models.

Besides Brandt et al. (2002) we have compared the FALL3D-8.0 model to more recent modelling of the Fukushima incident. However, since it's outside of the scope of the current paper we directly refer to Brandt et al. instead of "other models in the literature". The revised statement is:

"For the **Chernobyl-1986** test suite case, very good agreement between the model simulations and observations was found for the dispersal of radioactivity (i.e. $^{131}$I, $^{134}$Cs and $^{137}$Cs) resulting from the 22 April 1986 Chernobyl nuclear accident, consistent with the findings of Brandt et al. (2002)."

Line 456-460: This text is not relevant as the paper is not about model development. Further developments to the benchmark suite would be more appropriate.

Agreed. We have removed this statement and revised it to:

"Future developments of the test suite include adding more case studies, model inter-comparison studies against the validation datasets provided here and validation of probabilistic forecasts. In terms of model utilities, we plan to introduce the option of ensemble forecasts and to incorporate data assimilation."

Overall, I would suggest that there are too many figures.

Agreed. The total number of figures in the revised manuscript (not including Appendices) has been reduced from 18 to 10.

---

## Author Comment (AC2) · 10 Sep 2020

**Response to R2**

**R2 general comments**

The manuscript presents applications and evaluation of the Eulerian transport model FALL3D-v8, alongside a companion paper (Part 1), which presents the model physics and some limited verification. The applications are of SO2 and volcanic ash transport using a data insertion scheme, evaluated using satellite imagery; the transport and deposition of volcanic ash evaluated using tephra samples; and the transport and deposition of radionuclides evaluated using deposition measurements. This manuscript pertains to 'model evaluation', using the language of the journal. Quoting GMD's scope, they suggest that "where evaluation is very extensive, a separate paper focussed solely on this aspect may be submitted. . .typically, this comprises a comparison of the performance of different model configurations or parameterisations." In the case where the manuscript contains "substantial conclusions about geoscience rather than about models, and such papers are not suitable for submission to GMD." In its current form, in my opinion, the manuscript does not sufficiently evaluate the model to fit the scope of publication in GMD. Very little of the manuscript focuses on model evaluation – i.e. the model's ability to reproduce real world physics – with too much focus on the data used to evaluate. The paper has, however, passed the access review stage, suggesting that the topical editor has deemed the manuscript acceptable for GMD's scope. Therefore below I provide my suggestions for revisions to improve the manuscript, followed by technical comments.

We thank R2 for their review of our paper. Please find specific responses below to each point raised.

1. The manuscript does not sufficiently evaluate the model physics. There are many ways that this could be done in harmony to the companion paper. For example, one evaluation could be through simulation of an ash cloud using the emissions terms (1)-(4) in Sect. 3.2.3 detailed in the companion paper. An additional important evaluation case is the effect of including the fourth-order Runge-Kutta scheme in the solving scheme. The superiority of the new aggregation scheme in v8 over that used in v7 should also be demonstrated.

Most model physics parameterizations are inherited from previous versions and have been discussed and evaluated in several other papers (including sensitivity analyses). Moreover the fourth-order Runge-Kutta scheme was already evaluated in Part 1 (see Sect. 4 of the companion paper). In terms of model physics in the current manuscript, we evaluate the model's ability to simulate $SO_2$, long-range transported fine ash (using HAT and SUZUKI options), radionuclides and ash deposition. However, this is not the case of the new multi-class aggregation scheme, which we plan to consider in future studies.

2. The manuscript should include an example using emissions term (5) from Sect. 3.2.3 in the companion paper (i.e. resuspension). Desert dust would be a sensible choice if the authors wish to move the model away from being purely volcanological. This will better demonstrate dispersion from within the boundary layer.

Thank you for these suggestions. However, these emission schemes derived from mineral dust are already in v7.x and have been tested for volcanic ash resuspension events. See, for example:

- Mingari, L. A., Collini, E. A., Folch, A., Báez, W., Bustos, E., Osores, M. S., Reckziegel, F., Alexander, P., and Viramonte, J. G.: Numerical simulations of windblown dust over complex terrain: the Fiambalá Basin episode in June 2015, Atmospheric Chemistry and Physics, 17, 6759–6778, https://doi.org/10.5194/acp-17-6759-2017, 2017.
- Folch, A., Mingari, L., Osores, M. S., and Collini, E.: Modeling volcanic ash resuspension - application to the 14-18 October 2011 outbreak episode in Central Patagonia, Argentina, Nat. Hazards Earth Syst. Sci., 14, 119–133, https://doi.org/10.5194/nhess-14-119-2014, 2014.

3. A huge section of the paper is taken up by description of the satellite detection algorithms. This level of detail should not appear in the main text, which should focus on the model. No reasoning is given for using a bespoke satellite detection and retrieval algorithm here. A previously published algorithm should be used using an available data source to improve transparency. For example from SACS (https://sacs.aeronomie.be/) or some similar openly available source. This point is emphasised by the manuscript stating that (Line 321) 'it should be noted that the retrievals presented here are preliminary and require further cross-validation with other satellite retrievals'.

We agree with this reviewer comment. In response, we have removed almost entirely the satellite detection and retrieval sections from the main body of the manuscript. We now provide them as Appendices instead. We have added justification for using the previously published ash (Prata and Prata, 2012) and $SO_2$ algorithms (Prata et al., 2003) in new subsections in our revision. The relevant subsections are Sects. 3.1.1 and 3.2.1 of the revised manuscript. As is emphasised in our revision (and in our specific responses below), we note that SACS only provides polar-orbing $SO_2$ retrievals which do not fit our desired method of validation (high temporal resolution and large scale spatial coverage).

4. The choice of the 1986 Chernobyl accident seems an odd choice given the relative improvement in measurements during the Fukushima-Daiichi accident. This would also allow the authors to demonstrate the decay scheme for Strontium-90.

We agree with the reviewer that the Fukushima-Daiichi accident is a good choice for testing the model. We hope that the Fukushima case will be the subject of a future work. Here we decided to focus on the Chernobyl case since in that case the radionuclides were dispersed on a continental-scale area and a lot of measurements are available both at short and long distances.

5. It is unclear why there is so much emphasis on data insertion. The paper generally reads as justification for using data insertion, which has already been shown in Wilkins et al. (2016). Either the volcanic ash or SO2 example should be dropped as a single example shows that the model is capable of data insertion.

It was not our intention for this paper to be purely about data insertion. However, we concede that in its original form it could appear that way. To address this we have completely restructured the manuscript around the new FALL3D-8.0 test suite (as suggested by RC1). The new revised manuscript more concisely explains the reasons for selecting each validation case study (see Sections 3.1, 3.2, 3.3 and 3.4 of the revised manuscript) and how the various new features of the FALL3D-8.0 model have been tested. We also note that we have included several advancements based on the conclusions from the Wilkins et al. (2016) study, which include adding a source term to data insertion simulations (i.e. Puyehue-Cordon Caulle example), the first example of $SO_2$ data insertion (Raikoke example) and using CALIPSO lidar measurements to constrain the vertical distribution of the ash/$SO_2$ clouds.

**R2 technical comments**

Abstract: Acronyms (i.e. SAL/FMS) should not be defined in the abstract.
There is nothing wrong with defining an acronym in the abstract if it is referred to again within the abstract. Since we quote SAL and FMS scores in our revised abstract we have left the acronym definitions in.

Line 16: Change to '15+ year track record'
Done.

Section 2.1.: If keeping, a brief description of SEVIRI is needed.
Done. See Sect. 3.1.1 of the revised manuscript.

Section 2.1.1.: I would strongly urge the authors to use an 'off-the-shelf' product, but if keeping then it needs to be made clear that this is a bespoke algorithm relevant to this test case.
As far as we are aware, there are no "off-the-shelf" $SO_2$ retrieval products for geostationary satellites. Moreover, the method we are using has been published before and the previously published papers are already referred to in the text - so it is not a "bespoke algorithm" as the reviewer states. We have moved the specific assumptions used for our implementation of the 7.3 micron retrieval to the Appendix based on suggestions from RC1 and RC2.

Eq 1: The subscript 'ash' should not be italic and it would be clearer is Twc was replaced with -0.5K in the equation
We have changed subscript 'ash' so that it is not italic. However, we disagree with the reviewer's suggestion to replace the threshold definitions with constant values. Defining the thresholds as variables makes it much simpler to refer back to them in the text and in figures. For example, we quote the threshold settings in Figs. A1 and B2 (of the revised manuscript). It also makes it more straightforward to refer back to them in future studies if different threshold settings are used. In the end this comes down to style/preference. We also note that RC1 had no comment on these definitions.

Eq 3: Place -2K in the equation
See above comment.

Section 2.1.2, Eq 5: What geometric thickness of the cloud is assumed here? Is this the same thickness that is used in the insertion scheme?
This is an interesting question. We actually do not specify the geometric thickness of the ash cloud in the radiative transfer calculations as we are retrieving the total column mass. Eq. 5 (of the original manuscript) shows the relationship between the geometric thickness and the optical depth; however, in the radiative transfer modelling, we set the (total column) optical depth directly (from 0 to 9.9 in 0.1 increments) rather than compute it from the geometric thickness.

For the data insertion, we use the geometric thickness based on the CALIOP observation shown in Figure 4b of the original manuscript.

Eq 8: Put -2.5 K in equation
See above comment.

Line 172: Specify what you mean by 'meteorological clouds', i.e. water and ice clouds.
Done.

Line 178: "As mentioned above", specify the section.
We have re-worded this sentence (as the old subsections have been merged). See Appendix B of the revised manuscript:
"Considering that $\Delta T_{SO2}$ calculated via Eq. (B1) is a function of the total column density of $SO_2$, we can retrieve the total column amount by constructing this function from offline radiative transfer calculations."

Line 186: Which gases?
$H_2O$, $CO_2$, $O_3$, $N_2O$, CO and $CH_4$. These have been listed in the revised manuscript (see Appendix B).

Line 187: What 'amounts' are you referring to?
The total $SO_2$ column densities (in Dobson Units). This has been clarified in the revised text (see Appendix B).

Line 203: Is this vertical distribution also the same slab as used in the satellite retrieval? The Puyehue-Cordón Caulle eruption was known to have complex multi-layered cloud structures. How has this been dealt with in the satellite retrieval and insertion?
As mentioned above, we do not specify the ash layer thickness in ash retrieval. For the data insertion, we have selected a time when the ash layer was evidently uniform (i.e. almost representing a 'slab' in the vertical) in the CALIOP backscatter data (see Fig. A4b of the revised manuscript). Note that the vertical distribution of the ash layer only needs to be specified at the data insertion time.

Lines 212 and 216: FMS and SAL need defining in their first appearance in the mainbody of text.
Done.

Section 3: These validation metrics are only valid for the data insertion scheme. Please provide metrics for the other test cases. How S, A and L are combined into a single metric needs to be detailed.

How S, A and L are combined was detailed in the original manuscript at line 226-227.

However, to make his absolutely clear we have reworded the sentence to "After identifying objects for both the observation (satellite retrievals) and model fields, we compute the SAL as the sum of the absolute values of S, A and L, which results in an index that varies from 0 (best agreement) to 6 (worst agreement)". In addition, we have added the RMSE (along with its definition) as a quantitative validation metric for the other two case studies (Etna-2013 and Chernobyl-1986). See Sect. 4 of the revised manuscript for the definitions of the validation metrics.

Line 255: Are the 'ash mass loading areas' the areas of the satellite pixels or the meteorological/output resolution of the model? How are the alternate resolutions compared?

The ash mass loading areas are the areas of the model grid boxes. The satellite observations are first regridded (using nearest neighbour resampling) to the model grid. Then the areas where ash is present (above the defined threshold) in both the model and observations can be compared for spatial overlap. This was stated in the original manuscript on lines 200-203: "To insert IR satellite retrievals of volcanic ash and $SO_2$ (described in Sects. 2.1.2 and 2.2.2) into FALL3D, the satellite retrievals were re-sampled (using nearest neighbour sampling) from their native projection into a regular 0.1 X 0.1° latitude-longitude grid, consistent with the FALL3D grid." And reiterated in lines 227-228: "All comparisons between observations and model simulations are made using a regular 0.1 X 0.1° latitude-longitude grid". In our revised manuscript these details are provided in Sects. 3.1.1 and 3.2.1.

Sections 4.1 and 4.2: These sections are evaluating the data insertion method, which has been evaluated in previous work, rather than the model itself.

These sections evaluate the model's data insertion scheme *as well as* the model's emission sources without data insertion (e.g. HAT and SUZUKI options). The $SO_2$ simulation option (Section 4.2 of the original manuscript) has never been evaluated before so we're not sure what the reviewer is basing this assertion on. We hope that our revised and restructured manuscript has made this point clearer.

Section 4.1.: What is the grain size distribution used in this case? Assuming it is the retrieved effective radius, how many bins are used etc?

The simulation uses 12 particles bins (from 4 mm to 1 micron) assuming a particle size distribution that depends on column height (in this case this peaks at 125 microns). However, for validation against the satellite we only consider the 3 PM10 bins (sizes of 1, 4 and 8 microns) in order to be consistent with the range of effective radii retrieved by the satellite (1-10 microns).

Lines 259-263 and 264-269 can be cut. This information is superfluous to the model

We agree with the reviewer that this information is not relevant to this section of the paper. To address this we have re-organised the paper so that the description of the Puyehue eruption comes earlier on in the paper (Section 3) in a new "Validation cases" section (see Section 3.1 of the revised manuscript).

Line 299: Change 'meteorological clouds' to relevant cloud type

Done.

Lines 206-310 can be cut.

As stated above, we have restructured the paper so that the relevant pieces of information come earlier on in the manuscript. This comment suggests the complete removal of the description of the SAL metrics, discussion of previous authors that have used SAL and the description of the thresholds used to compare the model with observational data. As these pieces of information are crucial for understanding the validation, we have decided to retain this information in our revised manuscript.

Line 370: It is not clear to me why ARW was run first. Why was this initial step needed?

ARW was run first to generate the high resolution wind fields required to drive the FALL3D simulations. A more detailed explanation for using ARW can be found in our responses to RC1's technical corrections, response to Line 377 comment. Details of the ARW model configuration are also included in the revised manuscript Sect. 3.3.2.

Line 393: 'factor 3 error band' should just be 'a factor of 3'.

Done.

Line 399: 'nuclear accidents'

Done.

Line 414: In contrast to the lengthy explanation of the observations in the other test cases, nowhere in this section does it specify what was actually measured. This section also needs discussion on how general this set up is. For example, the FALL3Dv8 would be unable to be used to model the recent 2017 release of ruthenium-106 in Europe nor iodine-135/xenon-135 during Fukushima.

The referee is correct. As stated in Part 1 (Table 3), FALL3D-8.0 admits 5 different isotopes not including iodine-135/xenon-135. We plan to enlarge this list in future versions, including decay chains to other unstable elements. Regarding observations, high quality deposition measurements of particulates (i.e. radioactive isotopes) were measured and what is used in the present study for validation. We state this and also provide the validation dataset (as part of a GitLab repository) and a description of the model setup for the radionuclides simulations in the revised manuscript (see Sect. 3.4).

Line 416: The supplementary material contains nothing on how it accounts for diffusion, deposition nor decay.

We intended to refer to the radionuclide simulation animations that we provided as Supplementary Material. To avoid confusion we have removed the reference to Suppl. Mat. in the revised manuscript.

Section 5: Many of the conclusions are about the satellite detection scheme and applications of the model, rather than evaluation of the model itself.

Our revised conclusions now summarise the validation results for each case study (rather than specifics about the satellite retrievals). We also summarise next steps in terms of the test suite and future model inter-comparison studies. See response below.

Line 456-459: Model performance has not been discussed anywhere else in the manuscript and therefore does not serve as a conclusion/future work. This should be removed unless performance is explicitly detailed elsewhere in the manuscript.

We agree and have removed these statements. We have revised the concluding remarks so that they are based on the test suite, which is what this paper is about:

"Future developments of the test suite include adding more case studies, model inter-comparison studies that make use of the validation datasets provided here and validation of probabilistic forecasts. In terms of model utilities, we plan to introduce the option of ensemble forecasts and to incorporate data assimilation in future versions of FALL3D."

---

## Referee Report (RR1)

Thank you for considering my previous comment. Below I have some further comments/concerned that must be addressed before the manuscript is suitable for publication in GMD.

Considering my previous comment on the language of the comment, and the authors' response "In terms of model physics in the current manuscript, we evaluate the model's ability to simulate $SO_2$, long-range transported fine ash (using HAT and SUZUKI options), radionuclides and ash deposition", I suggest renaming the manuscript "FALL3D-8.0: a computational model for atmospheric transport and deposition of particles, aerosols and radionuclides – Part 2: model *evaluation*".
The language should be changed throughout the article to reflect this, e.g. the first sentence of the abstract, "This manuscript presents model evaluation…"

The sentence beginning on Line 93, starting "The eruption had multiple impacts…", has no relevance to the modelling and should be removed.

Sentence beginning line 131, starting "The International…", should be deleted as it contributes nothing to the paper.

Line 159: Which other $SO_2$ mass estimates? References are needed.

Line 160: "…these retrievals are preliminary and require more robust cross-validation…" These retrievals require more robust cross-validation in order to do what? Use them? This needs to be clarified.

Line 223: It is not clear what the work and findings of Brandt et al. is here – this needs stating if important.

Line 239: Use mm s$^{-1}$ to be consistent with the rest of the paper.

Line 246: "radioactive decay" should be changed to "the radionuclide half-life" as explicit radioactive decay (i.e. including daughter products) isn't demonstrated here.

Equation 3: RMSE, Mod and Obs should not be italicised. Else, choose a single letter/symbol to italicise. The variable j is not defined. The i of i-th on line 304 should be italicised. What are the values of $w_1$ and $w_2$?

Line 410: Write Bq m$^{-2}$ for consistency with the rest of the paper.

Throughout: consistency is needed whether "Mt" or "Mt." is written for e.g. Mt Etna.

Figures 4 and 5: Please replace the rainbow colour bar with a colour blind friendly alternative/more linear scale.

Figure 8: "Evaluated on" should be replaced (it wasn't done in 1986). E.g. Modelled total deposition accumulated until 10 May 1968.

Figure 9: Same as Fig. 8 for use of 'evaluated on'. 'Perfect coincidence' should be replaced with 'perfect agreement' or 'one-to-one agreement'.

Figure A4: Lidar images: Do not use a rainbow colour bar. Change to something more linear.

Figures B2 (d)-(f): Do not use a rainbow colour bar. Change to something more linear.

Figure B3: Use a linear/colour blind friendly colour bar on all plots.

Table 5: Change mm/s to mm$^{-1}$. Space is needed between number and units.

---

## Author Response (AR2)

**Response to R2 round 2**

**R2 general comments**
Thank you for considering my previous comment. Below I have some further comments/concerned that must be addressed before the manuscript is suitable for publication in GMD.
We thank R2 for their second review of our paper. Please find our responses below.

Considering my previous comment on the language of the comment, and the authors' response "In terms of model physics in the current manuscript, we evaluate the model's ability to simulate SO2, long-range transported fine ash (using HAT and SUZUKI options), radionuclides and ash deposition", I suggest renaming the manuscript "FALL3D-8.0: a computational model for atmospheric transport and deposition of particles, aerosols and radionuclides – Part 2: model evaluation".
The language should be changed throughout the article to reflect this, e.g. the first sentence of the abstract, "This manuscript presents model evaluation..."
Thank you for this suggestion. However, as the paper is about model validation and the term "evaluation" is not referred to in the manuscript, we have decided to keep the title in its current form. Quoting from the glossary of Jolliffe and Stephenson (2012): "In the meteorological context, forecast evaluation implies the study of forecast value involving user-specific losses rather than forecast quality, which is addressed in forecast verification". Therefore "evaluation" does not fit our intended meaning. The term "forecast verification" would be more precise, but it is synonymous with the term "forecast validation". Again from Jolliffe and Stephenson (2012): "forecast validation generally has the same meaning as forecast verification in meteorological forecasting". We also emphasise the fact that we have already changed the title once based on RC1's comments (please see our responses there for details). In addition, there are numerous examples of papers that have been published in GMD which use the term "validation" in the title: https://gmd.copernicus.org/articles/search.html?title=validation.

Jolliffe, I. T. and Stephenson, D. B. (Eds.). (2012). Forecast verification: a practitioner's guide in atmospheric science. John Wiley & Sons.

The sentence beginning on Line 93, starting "The eruption had multiple impacts...", has no relevance to the modelling and should be removed.
Accepted.

Sentence beginning line 131, starting "The International...", should be deleted as it contributes nothing to the paper.
We disagree. Referring the reader to a photograph taken from the ISS showing what the eruption plume actually looked like is very important for context and understanding the phenomenon we are modelling.

Line 159: Which other SO2 mass estimates? References are needed.

The SO$_2$ mass estimates that were referenced in the previous section *i.e.* from the Global Volcanism Program (2019) and Hyman and Pavolonis (2020). To clarify, we have added them at this line as well.

Line 160: "...these retrievals are preliminary and require more robust cross-validation..." These retrievals require more robust cross-validation in order to do what? Use them? This needs to be clarified.

Thanks. What we meant by this was that in order to provide a more detailed assessment of the retrieval uncertainties an error budget for the Himawari-8 instrument as well as cross-validation against other satellite retrievals would be required. Such a study is beyond the scope of the present study.

We have clarified this in the revised manuscript as follows:

"The present SO$_2$ retrieval scheme was originally developed for HIRS/2 (High-resolution Infrared Radiation Sounder) data. The error budget from Prata et *al.* (2003) suggests that errors from 10-20% are to be expected for detectable SO$_2$ column loads up to 800 DU. We expect that the Himawari-8 retrieval errors will be of similar magnitude or better".

Line 223: It is not clear what the work and findings of Brandt et al. is here – this needs stating if important.

Accepted. We have revised the statement to: "The availability of the Radioactivity Environmental Monitoring (REM) database (De Cort et al., 2007) along with the simulation results of Brandt et al. (2002) make the 1986 Chernobyl nuclear accident a good case study to validate the new radionuclide scheme in FALL3D-8.0".

Line 239: Use mm s -1 to be consistent with the rest of the paper.
Done.

Line 246: "radioactive decay" should be changed to "the radionuclide half-life" as explicit radioactive decay (i.e. including daughter products) isn't demonstrated here.

Radioactive decay is the physical process whereas "radionuclide half-life" is the time. We considered only the main radionuclides ($^{134}$Cs, $^{137}$Cs and $^{131}$I) for which the daughter products are all stable.

Equation 3: RMSE, Mod and Obs should not be italicised. Else, choose a single letter/symbol to italicise. The variable j is not defined. The i of i-th on line 304 should be italicised. What are the values of w 1 and w 2 ?

Thank you for these comments. We have removed italics for RMSE, Mod and Obs and i of i-th is now italicised. The variable j denotes the type of weight (j=1, 2), where $w_j$ refers to the weighting factor. We have clarified this in the revised text (see Equation 3 of revised manuscript).

Line 410: Write Bq m -2 for consistency with the rest of the paper.
Done.

Throughout: consistency is needed whether "Mt" or "Mt." is written for e.g. Mt Etna.
Thanks. Changed to "Mt. Etna" throughout.

Figures 4 and 5: Please replace the rainbow colour bar with a colour blind friendly alternative/more linear scale.
Done.

Figure 8: "Evaluated on" should be replaced (it wasn't done in 1986). E.g. Modelled total deposition accumulated until 10 May 1968.
Accepted. Removed the word "evaluated" to make the caption clearer.

Figure 9: Same as Fig. 8 for use of 'evaluated on'. 'Perfect coincidence' should be replaced with 'perfect agreement' or 'one-to-one agreement'.
Accepted. Removed "evaluated" and replaced "Perfect coincidence" with "one-to-one agreement".

Figure A4: Lidar images: Do not use a rainbow colour bar. Change to something more linear.
Total attenuated backscatter does not behave in a linear way. The NASA CALIPSO team specifically designed a colour scale to account for this. We have used the official CALIPSO colour scale instead.

Figures B2 (d)-(f): Do not use a rainbow colour bar. Change to something more linear.
Done.

Figure B3: Use a linear/colour blind friendly colour bar on all plots.
Done.

Table 5: Change mm/s to mm -1 . Space is needed between number and units.
Done.

[revised manuscript text omitted]